# Scaling Multi-Task Bayesian Optimization with Large Language Models

**Yimeng Zeng**[1]**, Natalie Maus**[1]**, Haydn Thomas Jones**[1]**, Jeffrey Tao**[1]**,**
**Fangping Wan**[2]**, Marcelo Der Torossian Torres**[2]**,**
**Cesar de la Fuente-Nunez**[2]**, Ryan Marcus**[1]**, Osbert Bastani**[1]**, Jacob R. Gardner**[1]
[1]Computer and Information Science, University of Pennsylvania
[2]Perelman School of Medicine, University of Pennsylvania
`yimengz@seas.upenn.edu`

## Abstract

In multi-task Bayesian optimization, the goal is to leverage experience from optimizing existing tasks to improve the efficiency of optimizing new ones. While approaches using multi-task Gaussian processes or deep kernel transfer exist, the performance improvement is marginal when scaling beyond a moderate number of tasks. We introduce `BOLT`, an initialization-only transfer strategy that distills prior BO runs into an LLM which proposes candidates for new tasks, while the surrogate at test time remains single-task. The LLM is periodically fine-tuned on top solutions from completed runs, creating a closed loop where better BO outputs yield better initializations over time. This decoupled design scales to roughly 1500 tasks without the saturation observed for shared-surrogate MTBO and adds only a small, amortized overhead relative to the BO inner loops. We evaluate on two domains: database query optimization and antimicrobial peptide design. We demonstrate that LLM-generated initializations steadily improve and accelerate BO, and with sufficient fine-tuning, a few LLM samples often match or surpass full "from scratch" BO with far fewer oracle calls.

## 1 Introduction

Multi-task optimization seeks to use related, previously observed tasks to accelerate the optimization of new ones. Multi-task optimization appears naturally in a variety of domains where similar problems are encountered repeatedly, such as hyperparameter optimization, material science, database query optimization, and drug design. Formally, suppose we have tasks $\{1, 2, \ldots, T\}$, each associated with its own objective function $f_t(\mathbf{x})$. For each task $t \in \{1, 2, \ldots, T\}$, we seek to find some $\mathbf{x}_t^*$ such that

$$\mathbf{x}_t^* = \arg\min_{\mathbf{x} \in \mathcal{X}} f_t(\mathbf{x}). \tag{1}$$

We focus on the setting where, for each task, we have collected a dataset $D_t$ of observations, and we wish to leverage this data when optimizing unseen test tasks.

Multi-task Bayesian optimization (BO) has traditionally learned across tasks by building a shared surrogate, typically via multi-output GPs and/or shared-weight neural feature extractors (Swersky et al., 2013; Perrone et al., 2018; Feurer, 2018; Patacchiola et al., 2020; Hakhamaneshi et al., 2022). A standard approach involves placing a multi-output GP over the input-task space, decomposing the kernel as an input kernel $k(\mathbf{x}, \mathbf{x}')$ and a task kernel $k(t, t')$. Despite their effectiveness, many of these methods — with the notable exception of recent work such as Wang et al. (2024b) — tend to saturate in performance after tens of training tasks and do not extract additional performance improvement on new tasks when given hundreds or thousands of related tasks.

We propose Bayesian Optimization with LLM Transfer (`BOLT`), a straightforward approach to multi-task BO that departs from the framework of building related task information into the BO surrogate model. Instead, as BO completes optimization for training tasks, we fine-tune a large language model (LLM) to, given a task description or context $C[f_t]$, generate solutions for that optimization problem that we can use as strong initialization for BO.

This approach creates a self-reinforcing feedback loop: BO generates high-quality solutions that we can leverage to fine-tune the LLM; the fine-tuned LLM, in turn, produces better initializations that improve BO performance. Over time, the LLM learns to directly generate solutions that are highly competitive, enabling top-$k$-samples from the LLM (requiring just a few oracle calls) to outperform full "from scratch" BO runs (requiring a large number of oracle calls). This iterative improvement enables `BOLT` to scale and still extract value from thousands of tasks. We validate `BOLT` on two diverse and challenging domains where many related tasks are available.

We transfer knowledge across tasks by *decoupling* it from the test-time surrogate and using it only for initialization. Rather than maintain a multi-task surrogate, `BOLT` distills prior experience into an LLM that proposes candidate solutions from a task description $C[f_t]$, after which a standard *single-task* BO run refines them. This removes shared-surrogate design choices, allows usage of any BO method unchanged, and improves with scale: as more tasks are solved, initialization quality rises rather than saturates. After sufficient fine-tuning on BO-discovered solutions, the LLM becomes a strong few-shot optimizer, and running BO on top of its samples yields further gains (see §4).

This design contrasts with recent LLM-based multi-task BO (MTBO) systems. Optformer seeks to predict entire optimization trajectories (Chen et al., 2022), and LLAMBO uses in-context surrogates with acquisition (Liu et al., 2024). `BOLT` instead uses the LLM strictly for initialization within a closed loop: BO finds high-quality solutions; we fine-tune on them; the LLM returns stronger starts. Ablations show that simple alternatives (e.g., sampling in trust regions around previous solutions) and an untuned LLM (`BOLT`-0) underperform, highlighting the benefit of the closed-loop, initialization-only approach (§4).

### Contributions

1. We propose `BOLT`, a scalable and *simple* alternative to traditional multi-task BO, leveraging LLMs to generate strong initial solutions for new tasks. `BOLT` leverages a combination of high-quality optimized solutions produced by BO and self-augmentation for fine-tuning.

2. We validate `BOLT` on two challenging, high-throughput domains—database query optimization and antimicrobial peptide design—and show that initialization quality *improves with scale*, avoiding the saturation of common shared-GP methods and outperforming recent LLM-based MTBO.

3. We show that, after sufficient fine-tuning, the LLM becomes a strong few-shot optimizer, often matching or surpassing full "from scratch" BO runs with far fewer oracle calls; running BO on top of those samples improves further.

4. We provide a detailed compute analysis and ablations demonstrating that `BOLT`'s fine-tuning/self-augmentation adds as little as $\sim 1\%$ overhead relative to single-task BO runs, adding minimal computational costs for extra performance.

## 2 BACKGROUND

**Bayesian optimization (BO).** Bayesian Optimization (BO) (Kushner, 1962; 1964; Močkus, 1975; Snoek et al., 2012) is an iterative approach to optimize black-box functions in a sample-efficient manner. On each step of the optimization, a supervised probabilistic *surrogate model* (usually a Gaussian Process (GP) (Rasmussen, 2003)) is conditioned on all data collected so far. Then, the surrogate model's predictive posterior distribution $p(y \mid \mathbf{x}, D)$ is used to decide what data point(s) should be evaluated next, typically by maximizing some *acquisition function*, defined with respect to $p(y \mid \mathbf{x}, D)$, which guides the exploration-exploitation trade off. Finally, selected points are evaluated on the black-box function and added to the dataset. This iterative process continues until the evaluation budget is reached.

**Structured optimization via latent space BO.** BO has recently been applied to optimizing structured search spaces, such as molecular and amino acid sequences, by leveraging latent space Bayesian optimization. This approach incorporates a variational autoencoder (VAE) to map structured inputs into a continuous latent space, where BO is performed (Kingma and Welling, 2014; Eissman et al., 2018; Tripp et al., 2020; Grosnit et al., 2021; Siivola et al., 2021; Stanton et al., 2022; Maus et al., 2022). Structured inputs $\mathbf{x}$ (e.g., amino acid sequences) are mapped to continuous latent representations $\mathbf{z}$ by the VAE encoder $\Phi(\mathbf{x})$. This creates a transformed continuous (latent) representation of the structured search space where BO can be directly applied (Gómez-Bombarelli et al., 2018; Griffiths and Hernández-Lobato, 2020; Kusner et al., 2017). The corresponding latent candidate points are then decoded by the VAE decoder, $\Gamma(\mathbf{z})$, to reconstruct structured outputs for

evaluation. For large combinatorial structured search spaces, such as the space of organic molecules or the space of all peptide amino acid sequences, the latent space of the VAE is typically high-dimensional (on the order of several hundred dimensions) in order to represent the large structured space effectively (Chu et al., 2024; Lee et al., 2025).

**Optimizing antimicrobial peptides.**  In antimicrobial peptide design, we seek peptides (sequences of amino acids) that minimize the MIC (minimum inhibitory concentration, measured in $\mu \, \text{mol} \, \text{L}^{-1}$) for some target bacterial pathogen. MIC is a measure of the concentration of the peptide required to inhibit growth of the target bacterial pathogen (Kowalska-Krochmal and Dudek-Wicher, 2021). A key challenge in antimicrobial peptide design is that many modern bacterial pathogens have developed resistance to modern antibiotics. To solve this challenge, Wan et al. (2024) propose designing new peptides with high sequence similarity to template peptides mined from extinct organisms. The template peptides themselves do not typically achieve sufficiently low MIC for target bacterial pathogens. However, since these template peptides have not been encountered in nature for thousands of years, modern antimicrobial resistant bacteria have not evolved resistance to them. It follows that new peptides are more likely to evade antibiotic resistance if they are designed to be similar to the extinct template sequences. We employ this strategy, optimizing antimicrobial peptides with a minimum threshold sequence similarity to the extinct template peptides from Wan et al. (2024). We also employ latent space BO to optimize over the structured space of amino acid sequences.

**Optimizing database query plans.**  Query optimization in data management systems involves translating a declarative SQL query into an execution plan that efficiently retrieves the correct results (Graefe and McKenna, 1993). This problem has been extensively investigated in the field of data management (Leis et al., 2017), as the difference in execution time between an optimal and a poorly chosen query plan can be several orders of magnitude (Leis et al., 2015). Since individual query plans are composed of discrete characteristics (e.g. join order trees), the search space of possible query plans is structured and combinatorial. We therefore employ latent space BO. We use the string representation for query plans proposed by Tao et al. (2025) to pre-train a VAE model that maps the structured space of query plans to a continuous latent space where BO can be applied.

**Database query plan optimization with right-censored observations.**  In database query optimization, our black-box objective function measures the execution latency of the query plan. "Good" and "bad" query plans can have latencies differing by multiple orders of magnitude (Leis et al., 2015). This can lead to the majority of optimization runtime being taken up by evaluating a small number of poorly performing plans. A natural solution to this problem is to *time out* objective function evaluations after they have reached some threshold latency $\tau$, resulting in *right-censored* observations. A right-censored observation is an observation at data point $\mathbf{x}$ where we observe only that $y \geq \tau$ for some chosen timeout threshold $\tau$, rather than observing the typical noisy objective value $y$. Prior work has been done to extend Bayesian optimization methods to the setting of right-censored observations. Hutter et al. (2013); Eggensperger et al. (2020) extended Bayesian optimization methods to the setting of right-censored observations by introducing an EM-like algorithm to impute the values of censored observations. Eggensperger et al. (2020) expanded on this, defining a single surrogate model capable of being conditioned on the combination of censored and uncensored data gathered. Tao et al. (2025) extend this to the setting of approximate GP surrogate models. Since we focus on tasks that involve large function evaluation budgets, we employ Tao et al. (2025)'s proposed method of modeling censored data with approximate GPs.

## 3 BAYESIAN OPTIMIZATION WITH LLM TRANSFER (BOLT)

We propose Bayesian Optimization with LLM Transfer (BOLT), an iterative framework for using LLMs to improve Bayesian optimization (BO) performance across a family of related tasks. We are given a set of $T$ training tasks defined by objective functions $f_1(\mathbf{x}), ..., f_T(\mathbf{x})$. We additionally assume that, for each objective function we have a *context* or *task description* $C[f_t]$ that can be a natural language or other input description that differentiates $f_t$ from any other task in the application domain. For example, this might be the text of a SQL query we are trying to optimize. Consequently, BOLT is specifically designed for domains with a shared input space and informative textual descriptions.

For each *training* task, we assume we have optimized the objective with some BO procedure, resulting in the optimization trajectories $\{\mathcal{D}_t^\star\}_{t=1}^T$, with each $\mathcal{D}_t^\star$ containing the top-$K$ observations from

the trajectory for the $t^{\text{th}}$ task. Our goal is to leverage this training data to learn an LLM-based "initialization policy" $\pi$ that, when presented with new related tasks $\{f_{T+1}(\mathbf{x}), C[f_{T+1}(\mathbf{x})]\}$, proposes a high-quality set of candidate solutions for BO to further refine.

These two procedures — (1) using BO to collect high-quality data for training tasks, and (2) using the LLM to initialize BO for new tasks — can be used as an "outer-loop"/"inner-loop" approach to solving a large number of related tasks sequentially, where the LLM is periodically updated as more optimization runs complete.

Because the LLM and BO only interact through generating initialization and generating fine-tuning data respectively, our approach here is relatively agnostic to the specific underlying implementation of BO used to optimize each task. This enables the straightforward use of the full range of recent BO advances on high-dimensional, constrained, and other optimization settings.

---

**Algorithm 1:** Inner Loop: LLM-Initialized Bayesian Optimization

---

**Require :** Task $t$, context $C[f_t]$, LLM $\pi_n$, budget $B$, batch $b$
**Ensure :** Optimized solutions $X_t^*$
$X_{\text{init}} \leftarrow \pi_n(C[f_t])$  // LLM proposes candidates
Evaluate $y_{\text{init}} \leftarrow f_t(X_{\text{init}})$
$\mathcal{D} \leftarrow (X_{\text{init}}, y_{\text{init}})$
Initialize $\mathcal{GP}(X_{\text{init}}, y_{\text{init}})$
**for** *step* $i = 1$ **to** $\lfloor B/b \rfloor$ **do**
  $X_{\text{next}} \leftarrow \arg\max_x \alpha(x; \mathcal{GP})$
   // Acquisition
  $y_{\text{next}} \leftarrow f_t(X_{\text{next}})$
  $\mathcal{D} \leftarrow (X \cup X_{\text{next}}, y \cup y_{\text{next}})$
  Update $\mathcal{GP}$ with new observations
Return $X_t^* \leftarrow$ top-$K(X)$  // Best solutions

---

**Algorithm 2:** Outer Loop: LLM Fine-Tuning via BO Trajectories

---

**Require :** Dataset $\mathcal{D}_0 = \{(C[f_t], \mathbf{x}_i, y_i)\}$, LLM $\pi_0$, iterations $T$
**Ensure :** Fine-tuned LLM $\pi_T$
Initialize $\mathcal{D} \leftarrow \mathcal{D}_0$, $\pi \leftarrow \pi_0$
**for** *iteration* $k = 1$ **to** $T$ **do**
  **foreach** *task* $t$ *in batch* **do**
    $X_t^* \leftarrow$ INNERLOOP$(t, \pi_k, B, b)$
     // Run BO
    $\mathcal{D} \leftarrow \mathcal{D} \cup \{(C[f_t], \mathbf{x}, y) \mid \mathbf{x} \in X_t^*\}$
       // Augment with top solutions
  Fine-tune $\pi_k$ on augmented dataset $\mathcal{D}$
  Update model parameters via instruction prompting
Return $\pi_T$  // Final fine-tuned LLM

---

**Initializing BOLT.** At initialization for a workload of tasks, we have only an un-tuned LLM BOLT-0 that is generally useless for the task setting because it is unaware of even the specific format for candidate suggestions. For the first iteration, we solve $T$ optimization tasks with a single-task BO routine where we initialize BO using some standard initialization procedure. We run optimization on each of the $T$ initial tasks, and extract the optimization trajectories $\{\mathcal{D}_i^*\}_{i=1}^T$ from each run.

**LLM fine-tuning.** The LLM fine-tuning process employs supervised learning using OpenAI's GPT-4O-MINI-0718 model through their API. From the optimization trajectories $\{\mathcal{D}_i^*\}_{i=1}^T$, we extract the top-$K$ observations from each of the $T$ runs completed so far. We use these observations along with the task contexts $\{C[f_t]\}_{t=1}^T$ to construct a fine-tuning dataset $\mathcal{D}_{\text{ft}}$. Each training instance contains:

1. A system prompt shared across all tasks in the workload/problem domain, which specifies the objective (e.g., generating efficient join orderings).

2. A user prompt with the task-specific context $C[f_t]$ (e.g., the SQL query requiring optimization).

3. A response prompt containing the high-performing solution $x$ discovered through BO.

We fine-tune using OpenAI's standard fine-tuning API (OpenAI et al., 2024). Specifically, we format our data into the required JSONL format (i.e., prompt-solution pairs) and then upload it via the fine-tuning API to initiate training. The model is trained to minimize the negative log-likelihood of the solution tokens $\mathbf{x}$ given the task context $C$:

$$\mathcal{L} = -\sum_{i=1}^{|\mathbf{x}|} \log \pi(x_i | C, \mathbf{x}_{<i}) \tag{2}$$

We note that our approach leverages full model fine-tuning rather than extensive and/or manual prompt engineering (Lester et al., 2021; Li and Liang, 2021). This allows the model to learn the task

requirements through the context-solution pairs in $\mathcal{D}_{\text{ft}}$, rather than explicit instructions. However, for scenarios requiring few-shot learning on untrained models, more careful prompt engineering may be beneficial. This fine-tuning process produces an updated model that encodes the knowledge from $\mathcal{D}_{\text{ft}}$. In our experiments, we will refer to an LLM trained on $T$ tasks in this way as BOLT-$T$.

**LLM fine-tuning frequency.** The number of tasks $T$ that we collect at initialization time and during each round of the BOLT "outer-loop" represents a non-trivial trade-off due to the computational cost of both running BO and the cost of fine-tuning the LLM. Fine-tuning the LLM more frequently results in both additional computational and monetary costs, but allows subsequent BO runs to complete more efficiently (with fewer black-box function evaluations). In this paper, we erred on the side of lower monetary cost in exchange for additional cost in black-box function evaluations. Specifically, we fine-tuned an LLM 4 times for the query plan optimization task and 7 times for the antimicrobial peptide design task as shown in Figure 2.

**Using the LLM for multi-task BO.** Once we have a fine-tuned model, BOLT-$T$, we can leverage the fine-tuned LLM's capabilities to generate higher-quality initialization points for subsequent optimization tasks. For a set of $n$ new tasks $\{t_i\}_{i=T+1}^{T+n}$, we sample from BOLT-$T$ to generate the same number of initialization points used by the baseline "from scratch" approach. The sampling prompt maintains the same structure as the training prompt without the assistant response. The BOLT-$T$ generated solutions are refined with a standard BO routine, and the top-$K$ performing solutions for each task $t$—along with their contexts $C[f_t]$—are incorporated into the training set for the next round of fine-tuning.

**Self-Augmentation.** As the fine-tuned LLM enhances few-shot generation with more optimization data, it is worth exploring whether the costly sequential BO processes can be minimized. Thus, we explore "self-improvement" methods to refine the LLM policy without the expense of additional optimization runs (Algorithm 3). Specifically, once an LLM has been fine-tuned using some of the tasks we set aside for training, we prompt it to generate additional solutions for *all* available tasks in that problem setting. We then score these solutions using the problem's oracle and fine-tune the LLM again directly with this labeled self-generated data, in a manner similar to self-instruction in LLM training (Shypula et al., 2024). By filtering and fine-tuning on its own best outputs, the LLM can iteratively teach itself how to propose better solutions.

---

**Algorithm 3:** Self-augmentation for LLM Finetuning

---

**Require :** Tasks $\mathcal{T}$, LLM $\pi_\theta$, iterations $T$, criteria $\mathcal{C}$
**Ensure :** Fine-tuned LLM $\pi_{\theta+T}$
Sample $\mathcal{D} \leftarrow \mathcal{D}_0$, $\pi \leftarrow \pi_0$
**for** *iteration* $k = 1$ *to* $T$ **do**
    **foreach** *task* $t \in \mathcal{T}$ **do**
        $X_{init} \sim \pi_{\theta+k}(t)$    // Generate samples
        $X_{init}^\star \leftarrow SelectBest(X_{init}, C)$
        // Select best samples
    $\mathcal{D} \leftarrow \mathcal{D} \cup \{(C[f_t], x, y) \,|\, x \in X_{init}^\star\}$
        // Augment dataset
    Fine-tune $\pi_{\theta+k}$ on $\mathcal{D}$
Return $\pi_T$   // Final fine-tuned LLM

---

## 4 EXPERIMENTS

We evaluate BOLT on two distinct problem domains, each with a large number of related tasks. For both domains, problem definitions and solutions can be represented as strings. This allows BOLT to operate both in sequence space, where the LLM learns from optimization trajectories, and in latent space, where BO makes additional progress using LLM-sampled initializations.

**Implementation details.** For the inner optimization loop, we implement a constrained version of the LOL-BO algorithm (Maus et al., 2022) using BoTorch and GPyTorch (Balandat et al., 2020; Gardner et al., 2018). For query optimization, we use an acquisition batch size of 1 with a budget of 4,000 oracle calls, while for peptide design, we employ a larger acquisition batch size of 50 with a budget of 20,000 oracle calls.

The outer loop BOLT-$T$ models use instruction prompting (Mishra et al., 2021; Longpre et al., 2023) to guide the LLM in producing optimized sequences. Figure 4 shows the template used to prompt GPT-4O-MINI for efficient query plans. After each optimization iteration, we augment the training set with the highest-scoring sequences from the optimization trajectory and fine-tune GPT-4O-MINI on this expanded dataset. When fine-tuning the LLM for the query plan optimization task, we use OpenAI's automatic batch size selection option. For the peptide design task, we found that using the

automatic batch size option did not provide a similar boost in performance, and we use a constant batch size of 10. For both tasks, we fine-tuned the LLM for 2 epochs and used the default OpenAI LR multiplier hyperparameter of 1.8. To ensure the solutions always have the correct syntax, we filter out characters that do not correspond to strings of integers or valid amino acids for the respective tasks. Figure 5 provides additional details on the fine-tuning process and prompts used for the peptide task.

**Database query plan optimization.** Database query plan optimization focuses on finding query plans (including join orderings and their operators) with low execution time for a given query. We take a subset of 2933 queries from the Cardinality Estimation Benchmark introduced by Negi et al. (2021), keeping 99 queries for validation. Following Tao et al. (2025), we perform BO over query plans by encoding join orders and operators as integer lists, which are then mapped to a 64-dimensional continuous latent space using the pre-trained query plan VAE from Tao et al. (2025). For pretraining, we randomly generated $1,169,890$ query plans based on the database schema, separated into 80/10/10 splits. For the initial "from scratch" runs with no LLM, we initialize with the set of 50 query plans used by BAO (Marcus et al., 2021), an ML-powered query optimizer we use as a baseline, that produces reasonable but non-optimal plans. Subsequent runs use 50 LLM-sampled query plans per query as initialization points. All points are sampled using a temperature parameter of 0.7 unless otherwise specified. The "task description context" $C$ used to fine-tune the LLM for this task is the full SQL query string; Appendix D shows that initialization performance is robust across nearby temperatures and that invalid generations remain rare in this regime.

**Antimicrobial peptide design.** For the peptide design application, we are given a library of 1000 extinct, weakly antimicrobial seed peptides $S = \{s_1, ..., s_L\}$. A task in this setting is to take a particular seed peptide $s_i$ and make modifications to it to minimize the minimum inhibitory concentration (MIC) against *A. Baumannii* ATCC 19606, measured in $\mu mol/L$. We created a library of $L = 1000$ extinct peptides and held out the last 100 as validation. We ensure edited peptides maintain a minimum 75% similarity to the seed peptide, defined by $1 - \frac{d(S,S')}{\text{len}(S)}$, where $d$ is the Levenshtein distance between them. All of the validation peptides are at least 25% different from any other peptide in the library. Although the seed peptides don't achieve low MICs, the hope is that bacteria are less likely to have developed resistance to their variations as they come from extinct species (Wan et al., 2024). We assess MICs with the APEX model and utilize a VAE trained on 4.5 million amino acid sequences to map peptides into a 256-dimensional latent space (Wan et al., 2024; Torres et al., 2024). Initial optimization uses 1000 randomly mutated sequences with a similarity constraint of 75% to the seed. Subsequent runs utilize 1000 LLM sampled peptides. All points are sampled using a temperature parameter of 1.0 unless otherwise specified. We use the seed amino acid sequence as the "task description context" $C$ for LLM fine-tuning.

**Baselines.** We compare `BOLT` against a range of baseline approaches. First, we compare to "from scratch," single task `LOL-BO` which we will refer to as `STBO`, which operates without prior task knowledge. Second, we compare to a common strategy for multi-task BO, e.g., Patacchiola et al. (2020); Hakhamaneshi et al. (2022); Perrone et al. (2018), where a shared GP is trained on all tasks through a neural network feature extractor using the optimization trajectories from training tasks. This shared GP is then checkpointed and used on test tasks. Several papers have found success with variations of this approach. ABLR (Perrone et al., 2018) uses independent Bayesian linear regression heads per task on the shared feature extractor, while FSBO (Wistuba and Grabocka, 2021) uses an adaptation of DKT (Patacchiola et al., 2020). In this paper, for the final supervised model, we use the same `PPGPR` model (Jankowiak et al., 2020) as the BO inner-loop in our method as we require scalability but find this results in better performance than Bayesian linear regression or random Fourier feature models (Rahimi and Recht, 2007).

We also compare against methods that utilize an ensemble of Gaussian process experts, POGPE and SGPE (Schilling et al., 2016). POGPE utilizes an ensemble approach with one expert from each previous task, while SGPE extends this by adding an additional expert trained on data for the current task only with higher weighting. We evaluate several configurations of these methods (with 5, 10, and 20 experts) to identify optimal performance. Additionally, we evaluate two transformer-based methods: Optformer (Chen et al., 2022) and LLAMBO (Liu et al., 2024). Optformer employs a fine-tuned transformer model (in our implementation, `GPT-4O-MINI`) for hyperparameter optimization, while LLAMBO uses out-of-the-box LLMs (also `GPT-4O-MINI` in our case, compared to `GPT-3.5` in the original paper) for both surrogate modeling and acquisition function optimization. Due to

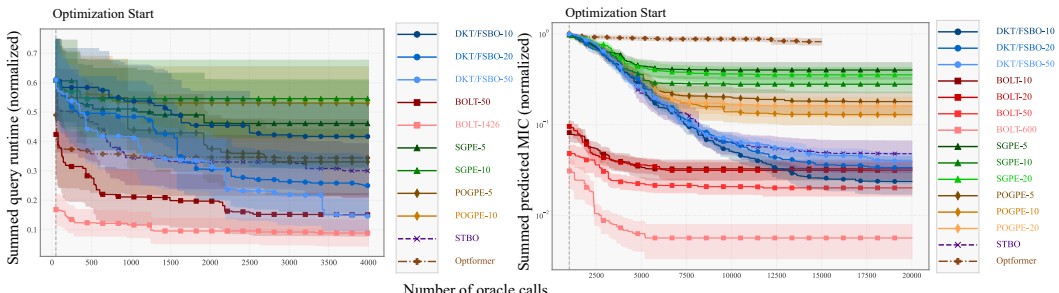

Figure 1: Bayesian optimization performance on **(Left)** query plan optimization and **(Right)** antimicrobial peptide design. The $x$-axis represents the number of oracle calls per held-out test task. In both settings, BOLT outperforms or matches baselines with just initialization data before optimization begins. BOLT-$T$ indicates that the LLM was fine-tuned on the top solutions from $T$ training tasks. Markers are plotted every 50 oracle calls, and lines represent the sum across test tasks, with shaded bands indicating $\pm 1$ standard error. Fine-tuning rounds for the LLM are *not* depicted on the $x$-axis: the LLM is updated only between batches of training tasks and remains fixed during evaluation on held-out tasks. At test time, each run starts from identically sized initialization sets (DB: 50 plans; peptides: 1,000 sequences), with BOLT replacing the baseline initializations with samples from the fine-tuned model. Consequently, the advantage observed at the "optimization start" reflects improved initial candidates, after which a standard STBO loop proceeds unchanged.

context length limitations with these transformer-based methods, we maintain a sliding window of the last 100 oracle calls for both approaches. For LLAMBO, we impose a budget limit of 10 million input tokens per experiment to manage computational costs. Additional details on all baselines can be found in Section C.

## 4.1 OPTIMIZATION RESULTS

In Figure 1, we demonstrate that initializing BO with BOLT significantly improves optimization efficiency across both domains. On the query optimization task (left), while DKT/FSBO makes improvements over STBO, the gains appear to plateau after only 20 tasks. In contrast, BOLT successfully scales to over 1400 tasks and converges to higher quality solutions faster. On the peptide design task (right), BOLT shows similarly strong performance, while DKT/FSBO struggles to take advantage of the data collected for separate templates. Notably, BOLT-generated initializations already outperform the respective baselines at the optimization start (i.e., before the first BO step on each test task), and this gap increases over the evaluation budget. We emphasize that the $x$-axis in Figure 1 records oracle calls made on the test task only; the LLM is not updated during these evaluations. All LLM fine-tuning occurs offline on previously solved training tasks, and in our runs, a small number of times per domain (four rounds for DB; seven rounds for peptides).

The GP expert methods show mixed results. For the database task, POGPE-5 and SGPE-5 demonstrate better performance than their variations with more experts, while for the peptide task, POGPE-10 and SGPE-10 yield the best results compared to 5/20 experts. Consistent with findings from Schilling et al. (2016), POGPE generally outperforms SGPE across both domains. However, both ensemble approaches are consistently outperformed by BOLT, and even fall behind STBO and DKT/FSBO in several cases. We did not run POGPE or SGPE with a larger numbers of experts as both methods scale poorly, requiring updating of a number of GPs proportional to the number of tasks.

The transformer-based methods demonstrate notable limitations in Figures 1 and 2. Optformer achieves performance slightly worse than STBO on the database task while showing significantly poorer results on the peptide task. LLAMBO performs substantially worse across both domains, showing minimal progress during optimization. Due to its computational demands—requiring LLM inference for both surrogate modeling and acquisition—LLAMBO completed fewer than 100 optimization steps within our token budget constraints.

We find that on both tasks once BOLT reaches a sufficient scale, it begins to few-shot generate initialization data for BO that is significantly better performing than the final results found by all baseline methods, including STBO, DKT/FSBO, the GP expert methods, and transformer-based approaches.

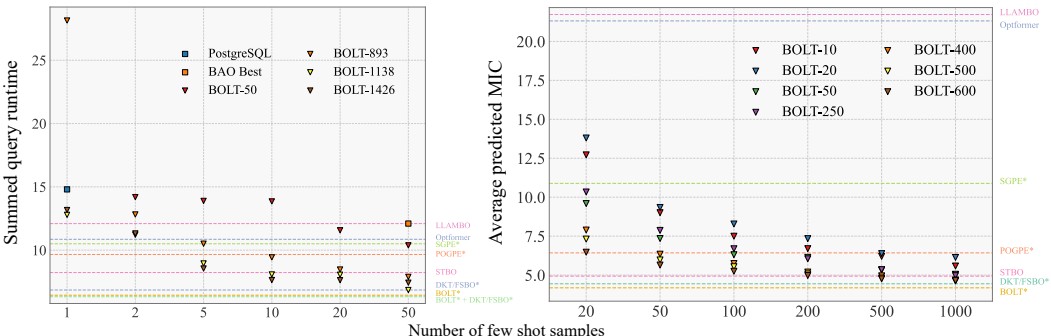

Figure 2: Evaluating BOLT in the few-shot setting and comparing to full optimization runs in both problem settings (**Left:** query plan optimization; **Right:** peptide design). In each plot, we show objective values accumulated across all validation tasks for various methods. Scatter points illustrate the few-shot performance of BOLT using different number of tasks, and relevant domain baselines (e.g., PostgreSQL, BAO for query optimization). Horizontal dashed lines indicate the performance of various full BO runs and other optimizers, shown for comparison. These results demonstrate that BOLT's few-shot performance is often comparable to or surpasses that of full BO runs.

**BOLT as a one- and few-shot optimizer.** In Figure 2, BOLT demonstrates strong few-shot generalization capabilities, even achieving single-shot performance competitive with traditional approaches. In query optimization, all BOLT variants outperform the top BAO solution within 5 samples. Notably, BOLT-1138 and BOLT-1426 surpass PostgreSQL in a single sample, indicating their potential for rapid deployment in low latency scenarios. The performance of BOLT consistently improves with more iterations across both tasks, except at 50 samples on the query optimization task, where BOLT-1138 slightly outperforms BOLT-1426. This may be due to variances in LLM sample generation or training. Overall, results confirm BOLT's robustness when scaling to thousands of tasks. We further compare our few-shot performance (Figure 2, before x-axis break) to full BO runs (Figure 2 after x-axis break). In both tasks, BOLT achieves few-shot results comparable to the full BO runs.

**Compute overhead and wall-clock.** Across both domains, the dominant cost is the inner-loop BO, not the LLM components. On the database tasks, an STBO run requires roughly 15–20 GPU-hours per task (avg. $\approx 18$), amounting to $\sim 25k$ GPU-hours over the full workload. By comparison, BOLT's outer-loop fine-tuning and sampling add only a small, amortized overhead. At the 50-task scale, the total compute with BOLT is $\sim$7% above STBO, and by the full workload of 1,426 tasks, it is $\sim$1% above STBO (Table 3). Fine-tuning consumed $\sim$60M tokens in total (OpenAI API; $\sim$\$180), which we also report as a conservative local-equivalent of $\sim$400 GPU-hours; generating 50 BOLT initializations per task took $\sim$1 GPU-minute per task ($\sim$24 GPU-hours across all DB tasks). Detailed per-method runtimes and token/cost accounting are summarized in Section B.

For completeness, we also quantify the cost of the self-augmentation step used in the outer loop: end-to-end, this adds on the order of tens of GPU-hours for the full workload (negligible relative to the BO inner loops); see Appendix for the precise accounting and setup.

## 4.2 ABLATION STUDIES

**LLM self-augmentation.** We investigate whether self-augmentation as outlined in Algorithm 3 can improve LLM performance while avoiding the computational expense of the inner-loop BO on the query plan optimization task. We apply the self-augmentation process to the four fine-tuned LLMs in Table 1, generating 10 samples from each across all $2,933$ training tasks, keeping only queries that outperform the best query plan from BAO's solutions. We then use these datasets as additional fine-tuning to create self-augmented versions of the LLMs.

Table 1 shows that this self-augmentation yields substantial improvements even without additional tasks optimized by BO. Both self-augmented models converge to a similar performance level, achieving a summed runtime of about 62 seconds across the 99 validation queries. This convergence suggests a natural performance plateau after training on either 1,500 tasks (BOLT-1426) or 1,100 tasks plus self-generated samples (BOLT-1138+SA). The consistency of this plateau across different training approaches further demonstrates BOLT's robustness when scaling to large task sets. This self-augmentation experiment indicates that once the LLM has been fine-tuned to suf-

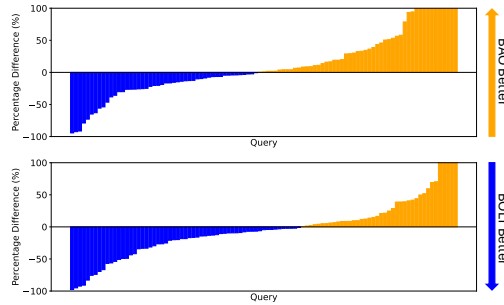

Figure 3: One-shot comparison of `BOLT-1426` to BAO (best-of-50). For each query, we sample one plan from the fine-tuned LLM and compare it with BAO's best-of-50 initialization (blue = BOLT better; orange = BAO better). Top: sampling with $T = 0.7$. Bottom: greedy with $T = 0.0$. In the one-shot setting, purely greedy sampling is better.

| Method | Best@50 |
|---|---|
| LLM (`BOLT-50`) no SA | 87.84 |
| LLM (`BOLT-893`) no SA | 82.31 |
| LLM (`BOLT-1138`) no SA | 78.16 |
| LLM (`BOLT-1426`) no SA | 63.68 |
| LLM (`BOLT-50`) | **82.25** |
| LLM (`BOLT-893`) | **63.05** |
| LLM (`BOLT-1138`) | **61.46** |
| LLM (`BOLT-1426`) | **61.54** |

Table 1: Ablation study for self-augmentation (SA) conducted on the query optimization task. For each of the LLMs with different training task sizes, we perform SA and generate 50 query plans from the LLM. We measure the best summed query execution time across the validation tasks from among these 50 samples.

ficient performance, it can generate additional fine-tuning data, reducing the number of BO runs required. Additionally, our framework scales to more training tasks without performance loss.

**One-shot capability.** Figure 3 isolates the one-shot behavior of the final, fine-tuned model: for each held-out SQL query, we draw a single plan from `BOLT-1426` and compare it to BAO's best-of-50 initial plans. No BO steps are run in this analysis. In particular, improvements to BO appear already with tens of training tasks (Figure 1), while the strong one-shot behavior in Figure 3 reflects the capability that emerges after scaling to many tasks.

**Impact of data quality on training.** We perform an ablation to assess the importance of using "better" versus "more" training data for fine-tuning LLMs through iterations of `BOLT`. Starting with the `BOLT-1138` model, we collect top solutions from a new BO round and train two variants: 1) `BOLT-1426`, which adds all new solutions to the original `BOLT-1138` set. 2) `BOLT-1138*`, which instead *replaces* an equal number of *old* solutions to maintain the same training set size. As shown in Table 2, both benefit from higher-quality data, suggesting "better" data boosts performance. However, `BOLT-1138*` underperforms `BOLT-1426`, which incorporates more and better data, confirming that both factors enhance model performance.

| Best@ | **BOLT-1138 no SA** | **BOLT-1138* no SA** | **BOLT-1426 no SA** |
|---|---|---|---|
| **Best@50** | 78.16 | 64.03 | 63.68 |
| **Best@20** | 82.59 | 70.52 | 66.23 |
| **Best@10** | 90.19 | 74.40 | 70.21 |
| **Best@5** | 102.99 | 85.21 | 76.28 |
| **Best@2** | 127.97 | 129.26 | 102.29 |
| **Best@1** | 202.04 | 193.64 | 160.22 |

Table 2: Comparing LLMs fine-tuned with **(Left)** data from 1138 tasks, **(Right)** data from 1426 tasks, and **(Middle)** data from 1138 tasks, but including the extra data from `BOLT-1426`, and removing data from older tasks. This is done on the DB task.

## 5 RELATED WORK

**Language models as optimizers.** Large language models (LLMs) have recently gained attention as sequence optimizers capable of tackling diverse black-box tasks where direct gradient information is unavailable or difficult to compute. LLM-based optimizers leverage the flexibility of natural language prompts to encode candidate solutions, constraints, and relevant task information. Methods like OPRO illustrate how iterative prompting can refine solutions (Yang et al., 2024; Zelikman et al.,

2024), while other approaches integrate self-improving strategies that reuse high-performing LLM outputs for further fine-tuning (Shypula et al., 2024). This set of techniques has been applied to biophysical domains such as molecular design and protein engineering, where the LLM proposes mutations to enhance certain properties, as well as to program optimization tasks where the LLM speeds up code execution time (Shypula et al., 2024; Wang et al., 2024a; Madani et al., 2023).

**Database optimization.**    Recent work has applied Bayesian optimization (BO) to improve overall database performance (Zhang et al., 2022; Nardi et al., 2019; Cereda et al., 2021) by tuning the parameters of the database configuration. As far as we are aware, Tao et al. (2025) were the first to apply BO to the specific setting of database query plan optimization considered in this paper. Other work has applied reinforcement learning (RL) to query plan optimization (Marcus et al., 2019; Yang et al., 2022; Zhu et al., 2023). RL query optimizers learn from mistakes and improve performance over time. Unlike BO, however, RL requires large supervised datasets for pre-training and typically aims to minimize cumulative query latency rather than achieving the lowest possible latency.

## 6    DISCUSSION AND LIMITATIONS

We first highlight a few limitations. First, our approach requires that all tasks in a problem setting have the same input domain (a problem that has been explored e.g. by Fan et al. (2022)). We further require the existence of a task description context $C[f_t]$ that can be used in an LLM prompt to define the task. This excludes common MTBO settings where tasks are primarily distinguished by data (e.g., hyperparameter optimization across datasets) rather than by concise textual descriptions; for such settings, approaches such as Wang et al. (2024b) are likely more appropriate. Finally, we note that the cost of LLM fine-tuning is significantly higher than simple gradient updates of a shared feature extractor, even though our experiments indicate that this overhead is small when amortized over thousands of tasks (§4, Appendix B).

Despite these limitations, in two real-world applications where BOLT was applicable it yielded strong results. Few-shot generation matched "from scratch" BO runs, and initializing BO from the LLM samples often improved performance further. Moreover, the interplay between the LLM and Bayesian optimization is noteworthy. Despite interest in using LLMs for optimization (Yang et al., 2024; Zelikman et al., 2024; Shypula et al., 2024; Wang et al., 2024a; Madani et al., 2023), finding initial strong solutions to fine-tune them is challenging in some domains. Bayesian optimization, by offering in-depth search, is an excellent candidate for this.

## REPRODUCIBILITY STATEMENT

We have released code at (`https://github.com/Yimeng-Zeng/BOLT`). We document task definitions, budgets (DB 4k calls; peptides batched at 50), and compute usage and hardware details. We report LLM fine-tuning hyperparameters and provide an open-source LLM recipe mirroring GPT-4O-MINI runs (see §4, Appendix A-D).

## ETHICS STATEMENT

This research introduces BOLT, a method leveraging large language models to enhance multi-task Bayesian optimization, with demonstrated applications in antimicrobial peptide design and database query optimization. The potential to accelerate the discovery of novel peptides could significantly benefit public health, particularly in combating antimicrobial resistance. Similarly, improving database query efficiency can lead to substantial computational and energy savings across many industries.

However, we acknowledge the potential for AI misuse in biological design. In applying these powerful methods, expert oversight, rigorous validation, and adherence to established safety and regulatory frameworks must be highlighted. Additionally, the use of large-scale LLMs raises considerations regarding computational accessibility and responsible AI development.

ACKNOWLEDGEMENTS

N. Maus was supported by the National Science Foundation Graduate Research Fellowship; J. R. Gardner was supported by NSF awards IIS-2145644 and DBI-2400135; C. de la Fuente-Nunez was supported by NIH grant R35GM138201, and by Defense Threat Reduction Agency grants HD-TRA11810041, HDTRA1-21-1-0014, and HDTRA1-23-1-0001.

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

## A    PROMPT DETAILS.

Figures 4 and 5 illustrate the prompt templates used for generating optimized query plans and peptide sequences, with GPT-4O-MINI-0718. Figure 4 shows the template for database query optimization, where the system acts as an assistant providing efficient join orderings for a given SQL query. Figure 5 displays the template for antimicrobial peptide design, where the system's role is to modify peptide sequences to enhance antimicrobial activity.

```
System: You are a helpful assistant that provides efficient join
    orderings for given queries.
User: {SQL query to be optimized}
Assistant: {Optimized query plan}
```

Figure 4: The prompt template used for prompting GPT-4O-MINI for generating optimized query plans.

```
System: You are a specialized assistant that modifies peptide
    sequences to enhance antimicrobial activity. Make up to 25%
    sequence modifications based on known antimicrobial peptide
    properties such as: positive charge, hydrophobicity, and
    amphipathicity.
User: {Seed peptide to be modified}
Assistant: {Modified peptide}
```

Figure 5: The prompt template used for prompting GPT-4O-MINI for generating optimized peptide sequences.

## B    COMPUTE DETAILS.

**Hardware.**    Experiments ran on an internal cluster (18 GPUs across two servers): one server with $8\times$ NVIDIA RTX A6000 (48 GB each; dual-socket CPU with 48 logical threads per socket) and one server with $10\times$ NVIDIA RTX A5000 (24 GB each; dual-socket CPU with 24 logical threads per socket).

**Scope.**    We report (i) per-task GPU-hours, (ii) aggregates at 50 tasks and 1,426 tasks, and (iii) LLM fine-tuning and inference usage (tokens, USD, and local GPU-hour equivalents). Database (DB) query-plan runs used one GPU per task unless specified.

RUNTIME AND COST OVERVIEW

**Single-task Bayesian Optimization (STBO).** Each DB run required 15–20 GPU-hours (avg. $\approx 18$). Across 1,426 tasks this is $\sim$25,000 GPU-hours, which is the dominant component of total compute.

**Baseline MTBO methods.**

- **SGPE/POGPE:** >100 GPU-hours per task; we capped runs at 110 GPU-hours. These approaches did not reach 1,000 oracle calls within reasonable time, and their cost increases quickly with the number of tasks.
- **DKT/FSBO (10/20/50):** 15–20 GPU-hours per task (similar to STBO). The extra feature-extractor cost is minor relative to GP training on the same data.

| Method | GPU-hours / task | GPU-hours / 50 tasks | GPU-hours / 1,426 tasks |
|---|---|---|---|
| STBO | 15–20 | ~900 | ~22,500–28,720 |
| SGPE/POGPE | >100 (capped at 110) | >5,000 (incomplete) | Infeasible (quadratic scaling) |
| DKT/FSBO | 15–20 | ~900 | ~22,500–28,720 (est.) |
| BOLT + STBO | — | ~964 (+7% vs. STBO) | ~22,750–28,970 (+1% vs. STBO) |

Table 3: **Runtime comparison on DB query-plan tasks.** BOLT's fine-tuning and inference add a *small, amortized* overhead relative to the BO inner loop: ~7% at 50 tasks and ~1% at 1,426 tasks.

**BOLT overhead (fine-tuning + inference).**

- **LLM fine-tuning (GPT-4O-MINI −0718 via API):** The largest run used 26M tokens (~\$78). Summed over BOLT-893/1138/1426, fine-tuning consumed 60M tokens (~\$180), roughly ~400 GPU-hours if performed locally on RTX A6000s (conservative equivalence).
- **Alternative local FT (Qwen 2.5–7B):** 32 GPU-hours total (4×A6000 for 8 hours).
- **Inference to generate initializations:** Sampling 50 candidates per task from a locally fine-tuned model took ~1 GPU-minute per task (about 24 GPU-hours across 1,426 tasks), negligible compared to BO.

**Takeaways.**   (i) The *vast majority* of compute is spent in the BO inner loop (STBO or comparable inner loops in MTBO baselines). (ii) BOLT's overhead—fine-tuning plus sampling a small batch of initial candidates—is small and amortizes quickly: ~7% at 50 tasks and only ~1% by 1,426 tasks (Table 3). (iii) SGPE/POGPE were substantially slower per task and did not scale to our full regime. (iv) Even when counting fine-tuning using a conservative *local* GPU-hour equivalent (~400 GPU-hours) rather than low-cost API usage (\$ ~180 total), BOLT's added compute remains marginal relative to the ~25k GPU-hours of BO.

**Token/API usage.**   Across BOLT-893/1138/1426, fine-tuning used 60M tokens (~\$180). Initialization inference across all DB tasks required ~24 GPU-hours in total.

## C   IMPLEMENTATION DETAILS.

### C.1   VAE TRAINING AND UPDATE DETAILS.

For optimization over structured input spaces, we utilize latent space Bayesian optimization via Variational Autoencoders (VAEs). The training and updating procedures for these VAEs differ by domain:

**Database query plans.**   For the DB task, we use a pre-trained query plan VAE (trained once on approximately 1.17M synthetic query plans generated from the schema). This VAE is not retrained or updated during our BO experiments.

**Antimicrobial peptides.**   For the peptide task, the VAE maps amino-acid sequences to a 256-dimensional latent space. Unlike the DB domain, this VAE is jointly updated with the surrogate model every 10 optimization steps. None of the $L = 1,000$ extinct seed peptides (900 training / 100 validation) used in our optimization tasks were included in the VAE's pre-training dataset.

### C.2   DKT/FSBO IMPLEMENTATION DETAILS.

For the antimicrobial peptide design task, a PPGPR model was trained using the GPyTorch module. This model employed a fully connected network with two hidden layers, each having a dimension of 256 and SiLU activations. Training parameters included a batch size of 128, a learning rate of 0.01, and 1024 inducing points for all peptide design experiments.

For the database query plan optimization task, the `PPGPR` model utilized a fully connected network with two hidden layers, each with a dimension of 64 and SiLU activations. A batch size of 16, a learning rate of 0.01, and 1024 inducing points were used for these experiments.

For both tasks, 50 STBO optimization trajectories were randomly selected. The DKT/FSBO-10/20/50 models were trained using the first 10, 20, or 50 of these trajectories, respectively. All models were trained for 20 epochs.

### C.3  POGPE/SGPE IMPLEMENTATION DETAILS.

Similarly to Section C.2, for the antimicrobial peptide design task, each expert model was a `PPGPR` model implemented with `GPyTorch`, with a fully connected network with two hidden layers, each with a dimension of 256 and SiLU activations. A batch size of 128, a learning rate of 0.01, and 1024 inducing points were used.

For the database query plan optimization task, each expert model uses a fully connected network with two hidden layers, each with a dimension of 64 and SiLU activations. The training used a batch size of 16, a learning rate of 0.01, and 1024 inducing points.

The same 50 STBO optimization trajectories from Section C.2 were used, and the first $5/10/20$ trajectories were used to train the POGPE/SGPE expert models. In POGPE, all experts were weighted equally. For SGPE experiments, the weighting scheme from Schilling et al. (2016) was adopted, where the independent GP for the target dataset carries the same weight as the entire set of experts.

### C.4  OPTFORMER IMPLEMENTATION DETAILS.

For both the query plan optimization and antimicrobial peptide design tasks, GPT-4O-MINI-0718 was fine-tuned on past optimization trajectories. To stay within context window limits, a maximum input context length of 100 trials and an output of 20 trials were used. The objective value ranges for both tasks were discretized into 1000 equidistant points. The training sets were constructed by randomly subsampling two trajectories of length 120 from the optimization trajectories. The query plan optimization task is trained on 27.4 million tokens and the antimicrobial peptide design task is trained on 4.8 million tokens. Both models were trained for 1 epoch with a batch size of 20 and an OpenAI learning rate multiplier of 1.8.

Optimization was initialized using the same points as single-task BO. During inference, a constant temperature of 0.7 was used. To manage inference token usage, a batch size of 20 was employed, where the model predicted the next 20 trials based on the previous 100 trials. This was important as experiments ran for 4,000 (query plan) or 20,000 (peptide design) trials.

### C.5  LLAMBO IMPLEMENTATION DETAILS.

The end-to-end LLAMBO method was utilized, leveraging GPT-4O-MINI-0718 for several components: generating candidate solutions, serving as a surrogate model for the objective function (via in-context learning), and acting as a conditional sampler to generate candidates for specific target values. Similar to Optformer, a maximum input context window of 100 trials was enforced to prevent exceeding context limits.

The hyperparameters from the original LLAMBO paper were adopted, including an exploration hyperparameter $\alpha = 0.1$, and $M = 20$. For the surrogate model, we sample $K = 10$ MC predictions to compute the empirical estimates. Consistent with the LLAMBO paper, we use the same sampling parameters with a temperature of $0.7$ and top_p of $0.95$. A limit of 10 million maximum input tokens per experiment was used to manage computational costs.

### C.6  LLM SELF-AUGMENTATION DETAILS.

For the antimicrobial peptide design task, 200 samples were generated for each of the initial $800$ training peptides during each self-augmentation round. Any peptides with a predicted MIC below 8 (indicating significant antimicrobial activity) were added to the training set for the subsequent round of `BOLT`.

For the database query plan optimization task, 10 samples were generated for each of the 2,933 training queries. Query plans with a runtime lower than the best plan generated by the BAO optimizer were added to the training set for the next round of BOLT.

## D  ADDITIONAL ABLATIONS.

### D.1  OPEN SOURCE LLMS.

To explore the viability of open-source models for BOLT, QWEN-2.5-7B and LLAMA-3.1-8B were fine-tuned using the identical dataset that created BOLT-1426 from GPT-4O-MINI-0718 for the database query optimization task. For evaluation, 50 query plans were generated from each LLM using a sampling temperature of 0.7. The best summed query execution time across the validation tasks from these 50 samples was compared.

Both models were fine-tuned on 4 NVIDIA RTX A6000 GPUs using a per-device batch size of 4, a learning rate of 1e-5 with the AdamW optimizer (Loshchilov and Hutter, 2019), and 5 training epochs. The results, shown in Table 4, indicate that QWEN-2.5-7B performed slightly worse than fine-tuned GPT-4O-MINI-0718, while LLAMA-3.1-8B showed significantly lower performance. Due to the extensive number of inference calls and multiple fine-tuning rounds required by BOLT, the primary experiments were conducted using the OpenAI API due to hardware resource limitations.

| Model | Summed runtime |
|---|---|
| GPT-4O-MINI-0718 | 61.54 |
| QWEN-2.5-7B | 62.04 |
| LLAMA-3.1-8B | 155.55 |

Table 4: Comparing open source LLMs fine-tuned with data used to fine-tune BOLT-1426 against OpenAI models fine-tuned on the same data.

### D.2  RANDOM PERTURBATIONS AROUND PRIOR SOLUTIONS.

We test whether small random perturbations around prior best solutions provide stronger initialization. In the latent space, we sample 50 candidates per validation task within axis-aligned trust regions (TR) centered at each prior best solution, with side-lengths $\ell \in \{0.5^7, 0.5^6, \ldots, 0.5^0\}$. We then decode and evaluate these candidates.

Table 5: **DB (first 10 validation queries).** Summed runtime (*seconds*; lower is better) when initializing from random perturbations of prior best solutions within latent-space trust regions. "Previous solutions" repeats Table 7 for reference. A large trust region can degrade the validity or quality of decoded candidates (marked with $^\dagger$).

| Method | Prev. sol. | TR $(0.5^7)$ | $(0.5^6)$ | $(0.5^5)$ | $(0.5^4)$ | $(0.5^3)$ | $(0.5^2)$ | $(0.5^1)^\dagger$ | $(0.5^0)$ |
|---|---|---|---|---|---|---|---|---|---|
| Summed runtime (s) | 9.18 | 9.18 | 8.94 | 8.94 | 8.92 | 8.97 | 9.07 | 38.48 | 8.62 |
| BOLT-1426 (init only) | | | | **7.43** | | | | | |

**Observation.** Local perturbations around prior best solutions offer modest gains over using the unperturbed pool (Table 5), but remain weaker than BOLT initializations. This indicates that task-conditioned sampling provides benefits beyond local neighborhood search around previous best solutions.

### D.3 COMPATIBILITY WITH GLOBAL BAYESIAN OPTIMIZATION

In our main experiments, we utilize local, trust-region-based Bayesian optimization algorithms, as they are well-suited for high-dimensional latent spaces. To demonstrate that BOLT is algorithm-agnostic and functions as a plug-and-play initialization module, we evaluate its compatibility with a standard global BO approach.

We implemented a recent global BO algorithm tailored for high dimensions (Hvarfner et al., 2024) within the same VAE latent space on the database query optimization task. We compared global BO initialized with the standard BAO baseline against global BO initialized with BOLT-1426 samples. As a reference, we also include our default local (trust-region) BO initialized with BOLT-1426. We evaluate performance up to 2,000 oracle calls rather than the full 4,000-call budget due to limited compute budget.

Table 6 reports the summed runtime on the first 10 validation queries. Before optimization begins, the baseline BAO initialization yields a summed runtime of 12.10 s, while the BOLT-1426 initialization starts at a much stronger 7.43 s.

Table 6: **DB (first 10 validation queries).** Summed runtime (*seconds*; lower is better) comparing global and local BO architectures. BOLT initializations substantially improve optimization trajectories regardless of the underlying BO algorithm.

| Method | 1,000 oracle calls | 2,000 oracle calls |
|---|---|---|
| Global BO + BAO init | 9.44 | 9.03 |
| Global BO + BOLT-1426 init | 6.94 | 6.92 |
| Local (trust-region) BO + BOLT-1426 init | **6.72** | **6.59** |

**Observation.** BOLT initializations substantially improve the performance of global BO (e.g., 9.03 s → 6.92 s at 2,000 calls). While local (trust-region) BO maintains a slight overall edge over global BO in this specific environment, the relative gain provided by BOLT remains consistent. This confirms that BOLT effectively identifies high-quality basins for new tasks, accelerating both global and local acquisition strategies.

### D.4 INITIALIZING WITH PRIOR BEST SOLUTIONS.

We test whether reusing the best solution from each previously optimized training task is a competitive generic initializer for new tasks. We collect the best-performing solution from every training task completed by BOLT-1426 and form a pool of "previous solutions." For evaluation, we consider the first 10 validation queries (DB domain), treat this pool as the initialization set (same size as other initializers), and measure the summed query runtime (lower is better). We compare against: (i) single-task BO (STBO) initialized with the standard baseline set; (ii) BOLT initializations from fine-tuned models with different training sizes; and (iii) a full BO run initialized by BOLT-1426. No further LLM fine-tuning occurs during this evaluation.

| Method | Summed runtime (s) |
|---|---|
| BOLT-1426 + BO | 6.43 |
| BOLT-1426 | 7.43 |
| BOLT-1138 | 8.65 |
| BOLT-893 | 9.08 |
| STBO | 8.23 |
| Previous solutions | 9.18 |

Table 7: **DB (first 10 validation queries).** Summed runtime (*seconds*; lower is better) under different initialization strategies. "BOLT-1426+BO" runs full BO after initializing with BOLT-1426 samples; other rows report initialization-only performance at the optimization start.

Simply reusing prior best solutions is less effective than model-generated initializations, and falls behind both STBO and all BOLT variants considered (Table 7). This suggests cross-task misalign-

ment: best solutions for earlier tasks do not align well with new tasks, while `BOLT` samples are tailored to the provided task description.

### D.5    SAMPLING ERROR RATES AND TEMPERATURE ABLATIONS

**Generation validity.**    We include a lightweight filter in our pipeline to remove characters that do not correspond to strings of integers (for DB plans) or valid amino acids (for peptides). This is primarily to ensure pipeline robustness, as non-valid samples are very rare at the sampling temperatures we use. To quantify this, we generated 50 samples for 100 DB tasks (5,000 total) across various temperatures $T$. As shown in Table 8, at our default DB temperature of $T = 0.7$, we generate 100% valid query plans. Only when pushing the temperature above 1.2 do we observe a noticeable increase in failed generations.

Table 8: **DB (validation set).** Generation validity across 5,000 samples per temperature setting.

| $T$ | Valid samples | Invalid samples | Valid (%) |
|-----|---------------|-----------------|-----------|
| 0.1 | 4,999 | 1 | 99.98 |
| 0.3 | 5,000 | 0 | 100.00 |
| 0.5 | 5,000 | 0 | 100.00 |
| 0.7 | 5,000 | 0 | 100.00 |
| 1.0 | 4,999 | 1 | 99.98 |
| 1.2 | 4,971 | 29 | 99.42 |
| 1.5 | 4,423 | 577 | 88.46 |

**Temperature sensitivity on DB.** We study the effect of the sampling temperature on initialization quality for the DB domain. For each temperature, we report the best-of-50 summed runtime across the standard validation set. We consider both GPT-4O-MINI-0718 and QWEN-2.5-7B, each fine-tuned on the same training data. As shown in Table 9, higher temperatures can slightly improve performance by increasing proposal diversity. We utilize $T = 0.7$, which is commonly used for LLM sampling and is close to optimal without affecting reported results.

Table 9: **DB (validation set).** Best-of-50 summed runtime (*seconds*; lower is better) vs. sampling temperature $T$.

| $T$ | GPT-4O-MINI-0718 | QWEN-2.5-7B |
|-----|------------------|-------------|
| 0.1 | 84.97 | 84.42 |
| 0.3 | 65.88 | 69.14 |
| 0.5 | 62.19 | 63.97 |
| 0.7 | 61.54 | 62.04 |
| 1.0 | 60.09 | 61.25 |
| 1.2 | 59.78 | 62.61 |
| 1.5 | 60.50 | 64.45 |

**Temperature sensitivity on peptides.** For the peptide task, we selected a slightly higher default temperature ($T = 1.0$) to increase the diversity of the 1,000 generated samples while maintaining adherence to the task's seed similarity constraint. Table 10 reports the uniqueness fraction, constraint-satisfaction fraction, and summed MIC as a function of $T$ (1,000 samples per seed; 20 validation seeds). $T = 1.0$ maximizes the effective fraction of peptides (both unique and within constraint) and yields the best summed MIC, confirming that it provides the best tradeoff between novelty and staying in-distribution.

Table 10: **Peptide temperature ablation.** For each sampling temperature $T$, we report the fraction of unique samples, the fraction satisfying the similarity constraint, the effective fraction (unique & in-constraint), and the summed best MIC across 20 validation seeds (lower is better).

| $T$ | Unique frac. | In-constraint frac. | Effective frac. | Summed best MIC $\downarrow$ |
|-----|--------------|---------------------|-----------------|------------------------------|
| 0.1 | 0.01620 | 0.87340 | 0.01285 | 155.91 |
| 0.3 | 0.12315 | 0.85800 | 0.09085 | 111.85 |
| 0.5 | 0.33190 | 0.83030 | 0.24170 | 103.94 |
| 0.7 | 0.56620 | 0.78585 | 0.40035 | 95.10 |
| 1.0 | 0.83990 | 0.69090 | **0.54350** | **93.30** |
| 1.2 | 0.93765 | 0.59700 | 0.53785 | 100.67 |
| 1.5 | 0.98655 | 0.39660 | 0.38425 | 103.56 |

### D.6 INITIALIZATION-STAGE COMPARISONS AND A NO-FINETUNING BASELINE

We report initialization only quality as a function of the first $k$ oracle calls on held-out tasks for both domains. We also include a no-finetuning LLM baseline (BOLT-0) in the DB setting.

**Peptides.** We show summed (unnormalized) predicted MIC across 20 validation peptides at $k \in \{1, 100, 200, 500, 1000\}$ oracle calls (lower is better).

Table 11: **Peptides (20 validation tasks).** Summed unnormalized MIC vs. oracle calls $k$ at the initialization stage.

| $k$ | BOLT-10 | BOLT–20 | BOLT-50 | BOLT-600 | STBO/MTBO |
|------|-----------|-----------|-----------|------------|-----------|
| 1 | 1204.0525 | 1120.9886 | 1057.1308 | **564.4510** | 5727.7421 |
| 100 | 135.4973 | 147.2456 | 112.9047 | **100.5810** | 1551.4898 |
| 200 | 120.4763 | 134.8256 | 107.1327 | **96.7752** | 1111.7161 |
| 500 | 109.1782 | 122.1278 | 101.3638 | **94.2405** | 792.5464 |
| 1000 | 107.5492 | 119.7096 | 97.1403 | **92.1007** | 625.9521 |

**Database queries.** We show summed runtime across 10 validation queries at $k \in \{1, 10, 20, 50\}$ oracle calls. We include BOLT-0 (no fine-tuning), BOLT-50, BOLT-1426, and STBO/MTBO.

Table 12: **DB (first 10 validation queries).** Summed runtime (*seconds*) vs. oracle calls $k$ at initialization. BOLT-0 uses an untuned LLM.

| $k$ | BOLT-0 | BOLT-50 | BOLT-1426 | STBO/MTBO |
|------|----------|----------|------------|-----------|
| 1 | — | 53.2584 | 13.9788 | 15.1161 |
| 10 | — | 13.8501 | 7.6380 | 13.7863 |
| 20 | — | 11.5664 | 7.6343 | 13.3761 |
| 50 | 182.9500 | 10.3764 | **7.4340** | 12.0967 |

The fine-tuned BOLT initializations improve markedly with scale and outperform both STBO initializations and the untuned BOLT-0 baseline early in the budget (Tables 11–12).

### D.7 ADDITIONAL OUTER-LOOP AND ROBUSTNESS ABLATIONS

**Outer-loop sensitivity on DB.** Table 13 reports Best@50 for three disjoint BOLT-200 runs, which lie between BOLT-50 and BOLT-893, indicating that the particular traces used at fixed $T$ matter less than the overall scale. Table 14 varies $K \in \{1, 2, 5, 10, 20\}$ at fixed $T = 893$ and shows better performance as $K$ increases.

Table 13: DB outer-loop sensitivity to which BO traces are used. Best@50 summed runtime (seconds; lower is better) across the 99 validation queries for three disjoint BOLT-200 runs and baselines (no self-augmentation).

| Model | Best@50 runtime (s) ↓ |
|-------|-----------------------|
| BOLT-50 | 87.84 |
| BOLT-200 (run 1) | 79.67 |
| BOLT-200 (run 2) | 84.09 |
| BOLT-200 (run 3) | 84.21 |
| BOLT-893 | 82.31 |

Table 14: DB outer-loop sensitivity to the number of top solutions per task. Best@50 summed runtime (seconds; lower is better) for `BOLT` trained on the same 893 tasks with varying $K$.

| Top-$K$ | Best@50 runtime (s) $\downarrow$ |
|---|---|
| 1 | 78.80 |
| 2 | 77.40 |
| 5 | 76.27 |
| 10 | 82.31 |
| 20 | 69.56 |

**Context shuffling on DB.** Table 15 shows that randomly shuffling the mapping between SQL contexts and `BOLT`-1426 plans degrades Best@50 from the original `BOLT` score to far worse than BAO, confirming that `BOLT` relies on aligned task descriptions rather than memorizing a global pool of good plans.

Table 15: DB context shuffling ablation. Best-of-50 summed runtime (seconds; lower is better) across validation queries when breaking the alignment between SQL contexts and `BOLT`-1426 plans.

| Method | Best-of-50 runtime (s) $\downarrow$ |
|---|---|
| BAO initialization | 106.72 |
| `BOLT`-1426 (aligned contexts) | 61.54 |
| `BOLT`-1426 (shuffled contexts) | 402.61 |

**Random and local-search baselines.** Table 16 summarizes random VAE-latent search, random query-space search, trust-region perturbations around prior best solutions, and local search around `BOLT`-1426 samples. Both random strategies and purely local perturbations underperform `BOLT` initialization, and even local search around `BOLT`-1426 remains weaker than running the full BO loop initialized from `BOLT`-1426, suggesting that task-conditioned proposals plus BO are essential.

Table 16: DB random and local-search-style baselines on the first 10 validation queries. Entries report summed runtime (seconds; lower is better) under comparable or larger oracle budgets.

| Method | Summed runtime (s) $\downarrow$ |
|---|---|
| `BOLT`-1426 + BO | 6.43 |
| `BOLT`-1426 + local search (latent perturbations) | 6.89 |
| `BOLT`-1426 init only (50 LLM samples) | 7.43 |
| TR perturbations around prior best solutions (50 samples) | 8.62 |
| Query-space random (4,000 samples, capped) | 9.42 |
| VAE-latent random (4,000 samples, capped) | 11.97 |

# E    LLM USAGE.

LLMs were fine-tuned for generating improved initialization points for BO runs as part of `BOLT`. LLMs were used to improve this paper's writing and presentation and assist in code implementation (e.g., co-pilot auto-completion). LLMs were not involved in generating or refining research ideas, experimental design, or theoretical developments.

