# OpenReview forum: "Scaling Multi-Task Bayesian Optimization with Large Language Models"
_ICLR.cc/2026/Conference — ICLR 2026 Poster_

### Official Review · Reviewer_PwEA · 2025-10-24

**Soundness:** 3
**Presentation:** 3
**Contribution:** 3
**Rating:** 6
**Confidence:** 4

**Summary:**

While this article claims to consider multi-task Bayesian optimisation, really it's about transfer learning within a class of Bayesian optimisation problems. The latter is really important, and the proposed solution is promising. The idea is to tune an LLM to map from task features to likely high performing parameter selections for that task, and to use these suggestions to initialise Bayesian optimisation when a new task comes along. The idea is good, and the results are interesting.

**Strengths:**

In the problem sets considered, the LLM is able to learn the strongly performing parameters from early runs of Bayesian optimisation. It learns so well that the initial suggestions for later tasks are often better than the final discoveries by single task BO on those tasks. However further refinement by BO provides even better solutions.

The avoidance of any need to learn a cross-task performance model makes multi-task BO much more feasible than in the traditional settings we consider.

**Weaknesses:**

The paper does not fully explore why this works. My intuition is that if you provide a few really strong examples to initialise BO, the acquisition function should then take the performance at these points as a baseline and go and search in the areas where it does not yet have any examples (ie far from the initialisation).

The number of BO reps is high. We are initialising with 50 observations and the x axis in Fig 1 goes to 2000 (so far as my eyes can make out). This is a lot of samples.

**Questions:**

Is my hypothesis correct on the consequence of initialising only with high value parameter values? If not, why not? Would it be better if we also transferred hyper parameters, or some other way of conveying the baseline performance level?

Given that we can find good initial points using the LLM, and apparently have access to a large number of samples, would local search not be likely to perform better? I'm not convinced that BO is the right approach once we have a well-fitting LLM.

How dependent is this on the particular BO method you have used? The BOSS algorithm (Moss et al, NeurIPS 2020) is explicitly designed for optimising over string spaces and would be a useful comparator. Is it that the *constrained* version of LOL-BO is really important (see my previous points). In which case, more should be made of it - perhaps an ablation where we use an unconstrained BO to demonstrate the difference.

Minor quibbles:
- I think you have a typo in line 177. You are not taking the argmax over alpha values. It should probably read argmax_x alpha(x,GP)?
- I disagree with your analogy to self-play in RL in line 243. In RL self-play we bring in new data from the environment when the algorithm plays itself. In self-instruction, and in what you do here, there is no further extrinsic information introduces at all.

---

> ### Author Response · Authors · 2025-11-21
> **Intuitions, initializations, and sampling budgets**
>
> **The paper does not fully explore why this works. My intuition is that if you provide a few really strong examples to initialise BO, the acquisition function should then take the performance at these points as a baseline and go and search in the areas where it does not yet have any examples (ie far from the initialisation).**
>
> In our setting, the inner-loop BO algorithms (LOL-BO / TuRBO variants) are explicitly local: they maintain trust regions centered at the current best points and adapt their size based on local "success" or "failure". Good initial points therefore do not simply act as a baseline that triggers the acquisition to jump far away, instead, they determine where the trust regions are centered and thus where most of the subsequent BO budget is spent. Intuitively:
>
> - BOLT provides high-quality, task-tailored points that are already in the basin of a good optimum.
> - The trust-region BO then refines these points locally, shrinking and expanding regions around them, rather than exploring arbitrarily far away.
> - Regions that are both far from the initial points and predicted to be bad tend not to have high acquisition values, even if they are uncertain.
>
> We will add a dedicated paragraph clarifying this interaction between BOLT and trust-region BO.
>
> **Is my hypothesis correct on the consequence of initialising only with high value parameter values? If not, why not? Would it be better if we also transferred hyper parameters, or some other way of conveying the baseline performance level?**
>
> Having high-value initial points is indeed important, and one of the main reasons BOLT works is that it reliably places BO inside the basin of good optima on new tasks. One important clarification is that BOLT is task-conditioned, not just general high-value parameter values. BOLT does not simply reuse globally good points or the top solutions from other tasks. Instead, the LLM conditions on the task description (SQL query or seed peptide) and generates high-value candidates tailored to that specific task. When we compare against “previous best solutions” reused directly as initializations, that baseline performs worse than both STBO and all BOLT variants (Table 6), indicating that task conditioning matters.
>
> Regarding transferring hyperparameters or baseline performance levels, this is exactly what our DKT/FSBO baselines do. They learn a shared feature extractor across different tasks, effectively transferring kernel hyperparameters and task-specific knowledge across tasks. In our experiments (which we will make clearer in the revised manuscript), we found that:
>
> - BOLT + STBO already improves substantially over STBO alone.
> - BOLT + DKT/FSBO improves further, indicating that parameter-level transfer and BOLT’s initialization-based strategy are complementary, not mutually exclusive.
>
> **The number of BO reps is high. We are initialising with 50 observations and the x axis in Fig 1 goes to 2000 (so far as my eyes can make out). This is a lot of samples.**
>
> These evaluation budgets are consistent with or lower than evaluation budgets seen in other high dimensional Bayesian optimization literature. For example, the TuRBO paper have a budget of 20,000 evaluations for 60D rover trajectory planning, and the LOL-BO paper runs out to 120,000 oracle calls for the Ranolazine MPO task.
>
> Our budgets of 4,000 evaluations for DB and 20,000 for peptides are in line with prior work on high-dimensional Bayesian optimization in similarly challenging domains.
>
> 1. TurBO https://arxiv.org/abs/1910.01739
> 2. LOL-BO https://arxiv.org/abs/2201.11872

---

> ### Author Response · Authors · 2025-11-21
> **Local search, BO algo, and other corrections**
>
> **Given that we can find good initial points using the LLM, and apparently have access to a large number of samples, would local search not be likely to perform better? I'm not convinced that BO is the right approach once we have a well-fitting LLM.**
>
> We agree that this is an important question and added local-search style baselines in Appendix D.3. We have
>
> - Random local perturbations around prior best solutions.
> In the DB domain, we construct latent-space trust regions around each previous best solution and sample 50 candidates per validation task at various side lengths. This gives moderate improvements over using the unperturbed pool (e.g., runtime improves from 9.18 s to 8.62 s for a carefully chosen region size), but still falls behind of BOLT-1426’s initialization-only performance (7.21 s).
>
> - Reusing prior best solutions directly.
> Simply initialising with the best solution from each training task ("previous solutions") is again clearly worse than BOLT (and even worse than STBO in our DB experiments).
>
> These ablations show that local neighbourhood search around earlier optima cannot explain BOLT’s performance. What makes BOLT better is that the LLM uses the task description to jump directly to high-quality regions for the new task, rather than just abusing existing solutions.
>
> Even once the LLM is strong, BO remains useful, we can see from Fig.2 that BOLT's few shot samples often match or surpass full from-scratch BO runs, but BO on top of those samples still yeilds additional non-trivial improvements. The BO surrogate in the high-dimensional latent space outperforms purely local perturbations.
>
> We will explicitly reference these local-search baselines in the main text (§4.2) and clarify that BOLT’s gains go beyond simple local optimization around previous or LLM-generated solutions.
>
> **How dependent is this on the particular BO method you have used? The BOSS algorithm (Moss et al, NeurIPS 2020) is explicitly designed for optimising over string spaces and would be a useful comparator. Is it that the constrained version of LOL-BO is really important (see my previous points). In which case, more should be made of it - perhaps an ablation where we use an unconstrained BO to demonstrate the difference.**
>
> Our framework is agnostic to the choice of inner-loop BO algorithm: BOLT only interacts with BO by (i) taking in trajectories of good solutions and (ii) returning an initialization pool. This is reflected in our experiments:
>
> - DB query optimization: we use a TurBO-style method with right-censored approximate GPs and no explicit constraints beyond the VAE decoder.
> - Peptide design: we use a constrained LOL-BO variant with SCBO to enforce $\ge 75\%$ sequence similarity to the seed peptides.
>
>
> Despite these different BO pipelines (constrained vs unconstrained, different latent dimensions, different objectives), BOLT consistently improves performance, and the pattern of improvement with scale is similar across both domains. This suggests that the choice of BO algorithm is not the main deciding factor of BOLT’s gains, instead, BOLT provides strong, task-aware initializations that any reasonable BO methods can utilize.
>
> BOSS is fully compatible with BOLT, we could simply treat BOLT's task-conditioned samples as initialization strings and let BOSS refine them, or use BOSS to generate trajectories that BOLT then learns from. Due to time and compute constraints, we were not able to implement and tune BOSS in our pipeline for the rebuttal, but we will make this plug-and-play nature explicit in the paper and note BOSS as a natural additional baseline/future addition for string optimization.
>
> **I think you have a typo in line 177. You are not taking the argmax over alpha values. It should probably read argmax_x alpha(x,GP)?**
> **I disagree with your analogy to self-play in RL in line 243. In RL self-play we bring in new data from the environment when the algorithm plays itself. In self-instruction, and in what you do here, there is no further extrinsic information introduces at all.**
>
> Thanks for pointing these out! We will update this in the revised manuscript.

---

> > ### Comment · Reviewer_PwEA · 2025-11-22
> >
> > This response raises two points.
> >
> > 1) Your described local-search-style baselines miss the point, in that they both start the search prior best solutions. Of course starting local search from previous high-scoring locations will not perform well. That's the whole point of BOLT; one needs to learn how to map context to good solutions to start STBO from a good point. I was attempting to ask about local search from the LLM-suggested start points? (If this is in the revised article but not in the comment above I apologize; my wife's parents are visiting this weekend and I prioritised getting the discussion started over waiting till they have left so that I have time to read the revised version.)
> >
> > 2) Your response on BOSS is not consistent with your previous answer when you claim that trust region / locally-biased formulations of BO are important. It would be really interesting to see what happens when a non-local BO method is used within BOLT.
> >
> > Both of these points are addressing my one real doubt about the article: once we have trained the LLM to predict good start points for the STBO, is STBO really the best way to do the within-task refinements?

---

> > > ### Author Response · Authors · 2025-11-26
> > >
> > > We apologize for the confusion in our earlier response, the most relevant comparison should be local search starting from the LLM-suggested candidates. To address this, we include a new baseline on the DB task, where for each of the first 10 validation queries, we take 50 samples from BOLT-1426, select the best 10 per task, and for each of these decode 400 local perturbations in the latent space (using the same perturbation scheme as in our “previous best solutions” ablation). Under the same oracle budget, pure BOLT-1426 few-shot initialization gives a summed runtime of **7.43s**, local sampling around the BOLT-1426 sequences improves this to **6.89s**, while BOLT-1426 + BO achieves **6.43s**. Thus, even when starting from the best LLM-generated initializations, BOLT + BO still provides a clear improvement over purely local sampling.
> > >
> > > For the BOSS comparison, we would like to clarify how strong initializations interact with global EI-based BO (which BOSS uses). Since EI is always computed relative to the current best value, once BO has seen a very strong point (as BOLT often provides), large portions of the space become effectively flattened out, where the surrogate predicts them to be clearly worse than the observed best, so their expected improvement is close to zero. The higher we push the best point, the region where EI is non-negligible shrinks, global EI then focuses most of its samples in the basin around the current best point. In this sense, even a global EI-based method behaves locally once a strong point is available. Empirically, in the high-dimensional spaces we consider, trust-region BO methods such as TuRBO and LOL-BO consistently outperform vanilla global BO runs in the same space [1], which is why we build BOLT on top of these local variants.
> > >
> > > To directly address your request for a non-local BO comparison, we are adding 1-2 additional baselines on the DB task using a standard global BO algorithm in the same latent space, both with random initialization and with BOLT‑1426 initializations. Based on the above intuition and prior work, we expect (i) global BO to generally perform worse than the trust‑region based methods in these high‑dimensional spaces, but (ii) within each BO method, replacing random initialization with BOLT initialization to still yield a clear improvement. We will post these “global‑BO + random vs. BOLT‑init” comparisons before the Dec. 3 discussion end date and also will include these in the revised manuscript.
> > >
> > > 1. [A survey and benchmark of high-dimensional Bayesian optimization of discrete sequences](https://proceedings.neurips.cc/paper_files/paper/2024/file/fe0007fcfd707673660ec0f9014bc48e-Paper-Datasets_and_Benchmarks_Track.pdf)

---

> > > > ### Author Response · Authors · 2025-12-03
> > > > **Additional global BO baseline**
> > > >
> > > > We have implemented a global BO algorithm (Hvarfner et al., 2024) in the same latent space on the DB task and compared it to our trust‑region local BO method, using the summed runtime metric on the first 10 eval tasks (lower is better). Due to time constraints, we only show comparisons up to 2000 oracle calls in this table, starting from the BAO baseline (12.10 s) and the BOLT‑1426 initialization (7.43 s), we get:
> > > >
> > > > | Method                                   | 1,000 oracle calls | 2,000 oracle calls |
> > > > | ---------------------------------------- | ----------- | ----------- |
> > > > | Global BO + BAO init                     | 9.44        | 9.03        |
> > > > | Global BO + BOLT‑1426 init               | 6.94        | 6.92        |
> > > > | Local (trust‑region) BO + BOLT‑1426 init | 6.72        | 6.59        |
> > > >
> > > > These results are consistent with our intuition that BOLT initializations substantially improve both global and local BO (e.g., 9.03 → 6.92 s under global BO), showing that BOLT is plug‑and‑play across different BO algorithms. Regarding BOSS, we tried to include it, but on our task the emukit CPU implementation for the string kernel was prohibitively slow if we want to scale to 4000 oracle calls, and the GPflow‑based GPU implementation is currently not documented well enough for us to make it work reliably before the rebuttal deadline. We will therefore not include the BOLT+BOSS experiment in our rebuttal.
> > > >
> > > > 1. [Vanilla Bayesian Optimization Performs Great in High Dimensions](https://arxiv.org/abs/2402.02229)

---

### Official Review · Reviewer_Yjw8 · 2025-10-28

**Soundness:** 2
**Presentation:** 3
**Contribution:** 3
**Rating:** 4
**Confidence:** 5

**Summary:**

&nbsp;

The authors introduce Bayesian optimization with LLM transfer (BOLT), a framework for multitask Bayesian optimization (MTBO) that leverages supervised finetuning (SFT) of LLMs to propose initial candidates for unseen tasks. The method is applied to database query planning and peptide design. Although the idea is interesting, I have concerns regarding the overall experiment design as well as the implementations of the baselines. Furthermore, I have concerns over the reproducibility of the work given that the codebase is not released. If these issues can be addressed in the rebuttal I will be willing to raise my score.

&nbsp;

**Strengths:**

&nbsp;

The method is novel and timely, leveraging advances in LLMs for multitask Bayesian optimization (MTBO). It is conceivable that LLMs can be strong meta learning tools for MTBO albeit the tradeoffs against existing methods in terms of performance and/or compute cost are unclear.

&nbsp;

**Weaknesses:**

&nbsp;

I summarize my comments in major and minor points. In particular points 2,3, and 4 are worth prioritizing for the authors to receive an upgraded score.

&nbsp;

**__MAJOR POINTS__**

&nbsp;

1. In terms of the structure of the paper, the background on the problem settings on page 3, namely antimicrobial peptide and database query plan optimization would be better placed in the appendix. Given that the authors are introducing a general purpose optimization methodology at a machine learning conference, readers are more likely to be interested in comparable optimization methods rather than the specifics of the applications considered. Given that the authors leverage LLMs, an expanded related work section on meta learning and/or transfer learning in BO as well as other attempts to use LLMs for BO may be more appropriate.

2. The absence of an anonymous GitHub or similar in the submission raises concerns for the reproducibility  of the current work e.g. the description of the architecture for the network in Section C.1 is not complete. What was the choice of activation function?

3. The problems considered by the authors feature optimization over structured input spaces via latent space BO. Following on from point 2 above some of the details of VAE training are missing. Do the authors periodically retrain the VAE? As I understand the authors use the same initialization set as the BAO algorithm for database query planning? Why not use random initializations and report errorbars? For the peptide problem the authors mention they curate L=1000 sequences partitioning 100 of these to the validation set. Is the VAE trained on the 900 sequences comprising the training set?

4. It would be worth adding a random search baseline across the VAE latent space as a sanity check.

&nbsp;

**__MINOR POINTS__**

&nbsp;

1. In terms of the opening statement, "Multi-task optimization seeks to use related, previously solved tasks to accelerate the optimization of new ones." I would disagree with this definition. Related tasks do not necessarily need to be "solved" to provide benefit to the optimization of a new task.

2. When citing domains in which multitask optimization problems occur, it would be useful to provide reference works for the domains mentioned.

3. Line 64, the acronym MTBO for multi-task Bayesian optimization is introduced before it is defined.

4. Line 64/64, the source papers for Optformer and LLAMBO should be given upon first mentioning the methods.

5. Line 86, when introducing Bayesian optimisation it would be worth citing the originating papers for the methodology [1,2] as discussed in [3] in place of the references given.

6. Line 89, the notation $y$ should be defined as a noise-corrupted version of $f(\mathbf{x})$.

7. The source paper for the VAE [4] should be cited when introducing it on line 96.

8. On line 97 when introducing the idea of latent space Bayesian optimization, the source paper should be cited [5], as well as close follow-up works on the topic [6, 7] which were published before Eissman et al.

9. Line 124, the citation to Leis et al. should be parenthetical e.g. (Leis et al. 2015) as opposed to narrative since the author's name is not part of the sentence. It would be worth correcting this across the manuscript.

10. When describing structured input spaces it would be worth explaining what the authors mean by "structured" as this may not be apparent to layreaders. Something akin to a "non-numerical, discrete input" may be appropriate together with examples such as images, molecules, or amino acid sequences.

11. Line 148, the acronym for LLMs has already been defined earlier in the manuscript.

12. In Algorithm 1, the notation $X_t^*$, $X_{init}$, $y$ etc. should be defined. $y_{init}$ should be denoted as a vector. Additionally, $y$ and $y_{next}$ should also be denoted as vectors for generality in the batch setting. On line 177, $x$ should also be a vector. $\alpha$ should not be a subscript on the argmax.

13. In Algorithm 2, it would be better not to use the variable $T$ for the number of iterations since it overloads the use of $T$ as the number of tasks. Additionally, $x$ and $y$ should be bolded as vector quantities $\mathbf{x}$ and $\mathbf{y}$.

14. Line 198, $x$ should be a vector.

15. There is a missing full stop at the end of Equation 2.

16. There are missing capitalizations in the references e.g. "bayesian" in place of "Bayesian".

17. There is a missing arXiv reference for Eggensperger et al. 2020.

18. There are missing conference references e.g. Haluptzok et al. was published at ICLR 2023.

19. It would help the reader if the authors described roughly what the DKT and FSBO methods were instead of forcing the reader to read the source papers.

20. In the related work on language models as optimizers it would be worth discussing the relation of the current work to [8].

21. The source paper for AdamW [9] should be cited given that it is used.

&nbsp;

**__REFERENCES__**

&nbsp;

[1] H.J. Kushner (1962). [A Versatile Stochastic Model of a Function of Unknown and Time Varying Form. Journal of Mathematical Analysis and Applications](https://www.sciencedirect.com/science/article/pii/0022247X62900112) 5(1):150–167.

[2] H.J. Kushner (1964). [A New Method of Locating the Maximum Point of an Arbitrary Multipeak Curve in the Presence of Noise.](https://asmedigitalcollection.asme.org/fluidsengineering/article-abstract/86/1/97/392213/A-New-Method-of-Locating-the-Maximum-Point-of-an?redirectedFrom=fulltext) Journal of Basic Engineering 86(1):97–106.

[3] Garnett, R., [Bayesian optimization](https://bayesoptbook.com/). Cambridge University Press. 2023.

[4] Kingma and Welling, [Auto-encoding Variational Bayes](https://openreview.net/forum?id=33X9fd2-9FyZd&source=post_page---------------------------), ICLR 2014.

[5] Gómez-Bombarelli, R., Wei, J.N., Duvenaud, D., Hernández-Lobato, J.M., Sánchez-Lengeling, B., Sheberla, D., Aguilera-Iparraguirre, J., Hirzel, T.D., Adams, R.P. and Aspuru-Guzik, A., 2018. [Automatic chemical design using a data-driven continuous representation of molecules](https://pubs.acs.org/doi/full/10.1021/acscentsci.7b00572). ACS Central Science, 4(2), pp.268-276.

[6] Griffiths, R.R. and Hernández-Lobato, J.M., 2020. [Constrained Bayesian optimization for automatic chemical design using variational autoencoders](https://pubs.rsc.org/en/content/articlehtml/2019/sc/c9sc04026a). Chemical Science, 11(2), pp.577-586.

[7] Kusner, M.J., Paige, B. and Hernández-Lobato, J.M., [Grammar variational autoencoder](https://proceedings.mlr.press/v70/kusner17a.html?ref=https://). ICML 2017. PMLR.

[8] Ranković, B. and Schwaller, P., 2025. [GOLLuM: Gaussian Process Optimized LLMs--Reframing LLM Finetuning through Bayesian Optimization](https://arxiv.org/abs/2504.06265). arXiv preprint arXiv:2504.06265.

[9] Loshchilov and Hutter, [Decoupled Weight Decay Regularization](https://openreview.net/forum?id=Bkg6RiCqY7), ICLR 2019.

&nbsp;

**Questions:**

&nbsp;

1. On line 155, the authors state that each training task comprises the top-K observations from the trajectory for each of $t$ tasks. Are these the top-K optimal observations or simply the top-K observations identified in a BO procedure. If the latter, how does one ensure consistency in the efficacy of the BO procedure used to generate the traces for each training task? Update: This is clarified to be the latter on page 4 of the paper. In this case it may be worth running a sensitivity analysis on the BO runs used to initialize the LLM with the top-K observations albeit this would be costly from a compute perspective.

2. In Figure 1, it is not clear how different tasks are treated. How does the number of oracle calls given in the x-axis related to the number of test tasks? What are the errorbars reported over?

3. I take it that BOLT-10, BOLT-20 etc. refer to the number of training tasks the LLM has been fine-tuned on? It might be worth emphasizing this for the reader in the experient section or the figure captions since the meaning of the BOLT-T notation appears only to be provided in Section 3. Update: This appears to be explained in the caption of Figure 2 but not in Figure 1.

4. For self-augmentation I fail to see the point of saving time on BO computation under the assumption that querying $f(\mathbf{x})$ (scoring under the problem's oracle in the authors' words) is a more expensive process relative to re-training the BO surrogate. Obviously the authors' BO runs were computationally intensive but is there an implicit assumption that for self-augmentation to be useful, evaluating $f(\mathbf{x})$ must be relatively cheap compared to the cost of running BO?

5. What is the task for the results reported in Table 4? The database task presumably?

&nbsp;

**Details Of Ethics Concerns:**

&nbsp;

No ethical concerns identified

&nbsp;

---

> ### Author Response · Authors · 2025-11-21
> **Background, code release, and implementation details**
>
> **In terms of the structure of the paper, the background on the problem settings on page 3, namely antimicrobial peptide and database query plan optimization would be better placed in the appendix. Given that the authors are introducing a general purpose optimization methodology at a machine learning conference, readers are more likely to be interested in comparable optimization methods rather than the specifics of the applications considered. Given that the authors leverage LLMs, an expanded related work section on meta learning and/or transfer learning in BO as well as other attempts to use LLMs for BO may be more appropriate.**
>
> In the revised manuscript, we will condense the background on antimicrobial peptide design and database query plan optimization in the main text, and move the more detailed domain descriptions to the appendix. We will also expand the related work section to more clearly position BOLT relative to (i) meta‑learning / transfer learning methods for BO (e.g., shared‑surrogate and pre‑trained GP approaches) and (ii) recent work using LLMs for optimization and MTBO.
>
> **The absence of an anonymous GitHub or similar in the submission raises concerns for the reproducibility of the current work e.g. the description of the architecture for the network in Section C.1 is not complete. What was the choice of activation function?**
>
> Thank you for raising the reproducibility concern, we agree it's an important part of open science! We are currently cleaning and packaging our code (inner-loop BO, outer loop fine-tuning and sampling scripts, dockerfiles) and will release it in an anonymous GitHub repository linked from a separate comment by this weekend. This will also make the remaining architectural details fully transparent. Concretely, the networks described in Section C.1 use two fully connected hidden layers (256 units for peptides, 64 for DB) with SiLU activations; we will add this missing detail to the appendix and ensure the repository reflects the exact configuration used in our experiments.
>
>
> **The problems considered by the authors feature optimization over structured input spaces via latent space BO. Following on from point 2 above some of the details of VAE training are missing. Do the authors periodically retrain the VAE? As I understand the authors use the same initialization set as the BAO algorithm for database query planning? Why not use random initializations and report errorbars? For the peptide problem the authors mention they curate L=1000 sequences partitioning 100 of these to the validation set. Is the VAE trained on the 900 sequences comprising the training set?**
>
> We apologize that the VAE setup was under‑specified in the current draft. For the DB task, we follow Tao et al. (2025) and use their pre‑trained query plan VAE, trained once on ≈1.17M synthetic query plans generated from the schema, this VAE is never retrained during our BO experiments. For the peptide task, we use a separate VAE trained on 4.5M amino‑acid sequences from UniRef/Torres et al. (2024), mapping peptides to a 256‑dimensional latent space, this VAE is jointly updated with the surrogate model every 10 optimization steps. None of the L = 1000 extinct seed peptides used in our optimization (900 train / 100 validation) are used to train this VAE.
>
> On initialization, for DB we start the first round of "from‑scratch" BO runs from the 50 BAO initial plans rather than random latent points, because BAO is an existing query optimizer that consistently provides better‑than‑random initializations. Below we added explicit random baselines (both VAE latent-space random and query‑space random) and showed that they perform substantially worse than BAO initializations and BOLT samples, confirming that this choice is conservative rather than cherry‑picking. For peptides, by contrast, there is no off‑the‑shelf optimizer tailored to our MIC objective, so we use random mutated sequences subject to a 75% similarity constraint as the baseline initializer.
>
> We will clarify all of the above in the experimental section and appendix (VAE training data, re-training/end-to-end updates, BO variants, and initialization strategy).

---

> ### Author Response · Authors · 2025-11-21
> **Random search, top-K, and sensitivity analysis**
>
> **It would be worth adding a random search baseline across the VAE latent space as a sanity check.**
>
> We have added two additional abalations on the DB task for a stronger random search baseline. First, we sample 4000 purely random points in the VAE latent space (~N(0,1)) for each of the first 10 validation queries and decode them, we excute the generated plans with a capped runtime that records any plan slower than the best BAO plan as having BAO's runtime, this will only make random search look stronger. Second, we add a query-space random baseline, using a QuickPick-style join-order generator with the same 4000 sample capped evaluation. We compare these agains (i) the local trust-region perturbations around prior best solutions from Appendix D.3 and (ii) the BOLT-1426 initialization only setting with just 50 LLM samples. On the first 10 validation queries, we have
> | Method                                          | Summed runtime (s) ↓ |
> |-------------------------------------------------|----------------------:|
> | BOLT‑1426 init only (50 LLM samples)            | 7.21                 |
> | TR perturbations around prior best solutions (50 samples)   | 8.62                 |
> | Query‑space random (4,000 samples, capped)      | 9.42                 |
> | VAE‑latent random (4,000 samples, capped)       | 11.97                |
>
> Here, we show that even optimistic random baselines that exhaust the full 4000 oracle call budget underperform both a simple local perturbation strategy, and BOLT's task-conditioned initialization with only 50 samples. We will add these results and a brief discussion to Appendix D.
>
>
> **On line 155, the authors state that each training task comprises the top-K observations from the trajectory for each of tasks. Are these the top-K optimal observations or simply the top-K observations identified in a BO procedure. If the latter, how does one ensure consistency in the efficacy of the BO procedure used to generate the traces for each training task? Update: This is clarified to be the latter on page 4 of the paper. In this case it may be worth running a sensitivity analysis on the BO runs used to initialize the LLM with the top-K observations albeit this would be costly from a compute perspective.**
>
>
> We confirm that the “top‑K observations” at line 155 are the top‑K points found by the BO trajectory for each training task, not globally optimal solutions. BOLT does not assume that every training run reaches the exact optimum, we let it learns from whatever high‑quality solutions BO discovers. To check sensitivity to which BO runs are used to initialize the LLM, we ran three non‑overlapping BOLT‑200 (no self-augmentation) variants on the DB task, each fine‑tuned on 200 disjoint training queries with K=10. Their Best@50 summed runtimes are 79.67, 84.09, and 84.21 seconds, compared to 87.84 for BOLT‑50 and 82.31 for BOLT‑893 (both without self‑augmentation; Table 1). With two runs lie between the original BOLT-50 and BOLT-893 curves, with one slightly outperforming BOLT-893 (no self-augmentation). This pattern is consistent with Figure 1's scaling trend and suggests that the variability across BO traces exists but is moderate relative to the performance gap between small-T and large-T BOLT runs. We will add these 3xBOLT‑200 ablations to the appendix and clarify in the main text that we fine‑tune on top‑K points from individual BO runs.
>
> We also include an abalation on the specific choice of K. Keeping the same training set as the original BOLT‑893 run, we fine‑tuned four models using $K\in{1,2,5,20}$ top solutions per task. Their Best@50 summed runtimes are shown in the table below.
> | Top-K | Best@50 summed runtime (s) ↓ |
> |------:|------------------------------:|
> | 1     | 78.80                         |
> | 2     | 77.40                         |
> | 5     | 76.27                         |
> | **(original BOLT-893) 10** | **82.31** |
> | 20    | 69.56                         |
>
> This shows that (i) BOLT's performance improves as we increase the amount of high-quality data, and (ii) our original K=10 choice was rather on the conservative side rather than over-tuned, as larger K values showed better performance at the cost of more fine-tuning tokens (16 million tokens for K=20). We will report these results in the appendix and note in the main text that outer-loop hyperparameters such as K can shift performance by a few seconds but do not change the overall scaling behavior or the conclusions of Figure 1.

---

> ### Author Response · Authors · 2025-11-21
> **Self augmentation and other clarifications**
>
> **For self-augmentation I fail to see the point of saving time on BO computation under the assumption that querying (scoring under the problem's oracle in the authors' words) is a more expensive process relative to re-training the BO surrogate. Obviously the authors' BO runs were computationally intensive but is there an implicit assumption that for self-augmentation to be useful, evaluating must be relatively cheap compared to the cost of running BO?**
>
> For self-augmentation, take the DB task as an example, we generate only 10 extra samples per query across 2,933 tasks (29330 additional oracle calls). This is roughly the equivalent of 7 extra full BO runs at a 4,000 call budget. In return, we substantially improve the LLM's performance. For example, BOLT-50 improves from 87.84s -> 82.25s (Best@50, Table 1), matching the performance of BOLT-893 at 82.31s (no self-ugmentation), achieving the improvement of ~840 additional full BO runs, which is around ~14000 hours of GPU compute (Appendix B, Table 2, A5000/6000 equivalent GPUs). So the point of self-augmentation is not that oracle calls are cheaper than retraining the surrogate/running BO, but that once we have a reasonably good LLM, we can "upgrade" it using the equivalent of only a couple of extra BO runs instead of hundreds, making it attractive whenever full BO runs are the main computational bottleneck. We will clarify this in the text around Table 1.
>
> **In Figure 1, it is not clear how different tasks are treated. How does the number of oracle calls given in the x-axis related to the number of test tasks? What are the errorbars reported over?**
>
> We apologize for the lack of clarity. In Figure 1, the x‑axis is the number of oracle calls per test task. For each method, we run every DB experiment to 4,000 calls and every peptide experiment to 20,000 calls, independent of the number of test tasks. At each x‑tick, we compute the normalized sum over all test tasks (summed normalized query runtime or summed normalized predicted MIC) and plot the mean, with error bars showing ±1 standard error across test tasks. All curves are evaluated on the same held‑out tasks; the only difference between lines is the BO/initialization method.
>
>
> **I take it that BOLT-10, BOLT-20 etc. refer to the number of training tasks the LLM has been fine-tuned on? It might be worth emphasizing this for the reader in the experient section or the figure captions since the meaning of the BOLT-T notation appears only to be provided in Section 3. Update: This appears to be explained in the caption of Figure 2 but not in Figure 1.**
>
> Yes, BOLT-10,20,...,T refer to the LLM fine-tuned on the top-K solutions from T training tasks (The number of BO runs the LLM has seen). In the revised version we will also explicitly define this in the Figure 1 caption and experiments section to make it clearer to the readers.
>
> **What is the task for the results reported in Table 4? The database task presumably?**
>
> Yes, Table 4 reports results for the DB task. We will update the table caption and the surrounding text to make this clear.

---

> ### Author Response · Authors · 2025-11-24
> **Code release**
>
> We have uploaded our code to https://anonymous.4open.science/r/BOLT-anonymous-release-20B6 for review

---

### Official Review · Reviewer_KGBF · 2025-11-01

**Soundness:** 3
**Presentation:** 3
**Contribution:** 3
**Rating:** 6
**Confidence:** 3

**Summary:**

The paper presents BOLT, a multi-task Bayesian optimization method that transfers information between tasks by fine-tuning a language model. Unlike other approaches, information transfer is done at the initialization level, such that initial values for the Bayesian Optimization procedure are chosen increasingly effectively as information from other tasks are incorporated into the language model. The authors demonstrate substantial improvements over earlier methods - in particular, they show better scaling with the number of tasks.

**Strengths:**

**Originality**. To paper introduces a novel way to use language models to improvement multi-task BayesOpt.

**Quality**. The method is described in detail, the empirical results support the stated claims.

**Clarity**. The paper is well-written, clearly explaining the idea. A reproducibility statement is included to clearly state what code will be made available upon acceptance.

**Significance** The paper reports significant improvements over the state of the art. The method is also quite simple, and is thus likely to find real-world application.

**Weaknesses:**

Although the paper presents several succesful applications, it does not provide a clear picture of when it fails. Limitations are discussed in broad terms, but the paper would benefit from e.g. a more careful analysis of how sensitive the performance is to the similarity between tasks. For example, one could imagine an experiment on the peptide design task where performance was reported as a function of the degree of similarity between peptides, to see if it broke down with very low similarities.

As is explicitly stated in the paper, the method focuses on large-data domains. However, it remains unclear how much data is typically necessary, and whether the approach could find application in small-data regimes with more restricted fine-tuning procedures. Any insights on this matter would be useful for a reader that is considering implementing the method.

**Questions:**

### Questions
line 190. *"we extract the top-K observations from each of the T runs completed so far."* Don't you risk getting very similar solutions using this approach. Wouldn't it make sense to impose some diversity criterion?

line 268. *"we filter out characters that do not correspond to strings of integers or valid amino acids for the respective tasks."*. How frequently do such non-valid sample errors occur? It this a problem in practice?

line 280. *"l points are sampled using a temperature parameter of 0.7 unless otherwise specified."*. For the two problems you use two different temperature hyperparameters. How are these tuned, and is the performance very sensitive to this choice?

### Minor comments

line 257. I found it difficult to assess whether a budget of 200,000 oracle calls is reasonable in practice. You could consider adding a note on this.

Figure 1. The labels in this figure are difficult to read, both due to the colours and the small font. Considering changing this.

---

> ### Author Response · Authors · 2025-11-21
> **Failure modes and dataset sizes**
>
> **Although the paper presents several succesful applications, it does not provide a clear picture of when it fails. Limitations are discussed in broad terms, but the paper would benefit from e.g. a more careful analysis of how sensitive the performance is to the similarity between tasks. For example, one could imagine an experiment on the peptide design task where performance was reported as a function of the degree of similarity between peptides, to see if it broke down with very low similarities.**
>
> We appreciate the suggestion to more clearly characterize failure modes and sensitivity to task similarity. In the current peptide setup, validation seeds are already constrained to be at least 25% different from any other peptide in the library, but we do not yet present results as a function of finer‑grained similarity bands. In the revised manuscript we will expand the limitations section to explicitly note that BOLT may degrade when tasks are only weakly related. We will also add a brief analysis for the peptide domain that categorizes the performance by seed similarity.
>
> **As is explicitly stated in the paper, the method focuses on large-data domains. However, it remains unclear how much data is typically necessary, and whether the approach could find application in small-data regimes with more restricted fine-tuning procedures. Any insights on this matter would be useful for a reader that is considering implementing the method.**
>
> Our main goal is large-data settings, where BOLT's fine-tuning cost amortizes over many tasks. Though we do see early gains with limited data. In peptides, BOLT-10/20/50 already improves initialization quality over STBO at the very start of optimization (in Appendix, Table 8). In databases, BOLT-50 reduces summed runtime at the initialization stahe by ~16% within the first 50 oracle calls (Table 9). For even smaller number of tasks (<10), traditional shared-surrogate methods (DKT/FSBO) are reasonable approaches. However, we do show that they plateau after ~20 tasks, whereas BOLT continues to improve as data scale (Figure 1). We will add a clarifying paragraph summarizing these and recommending DKT/FSBO when only a handful of tasks are available, and highlight that BOLT is the better choice once tens to hundreds of tasks are present.

---

> ### Author Response · Authors · 2025-11-21
> **Sampling error rates and temperature abalations**
>
> **line 268. "we filter out characters that do not correspond to strings of integers or valid amino acids for the respective tasks.". How frequently do such non-valid sample errors occur? It this a problem in practice?**
>
> Non-valid samples are very rare at the sampling temperatures we use. In the DB experiments our default sampling temperature is T=0.7, at which we generate 100% valid query plans. We further evaluate this on sampling temperatures of T = \{0.1/0.3/0.5/0.7/1.0/1.2/1.5\}, where we generate 50 samples for each of the 100 validation tasks, and evaluate if the sample is a valid string.
>
> | Temperature | Valid samples | Invalid samples | Valid (%) |
> | ----------: | ------------: | --------------: | --------: |
> |         0.1 |         4,999 |               1 |     99.98 |
> |         0.3 |         5,000 |               0 |    100.00 |
> |         0.5 |         5,000 |               0 |    100.00 |
> |         0.7 |         5,000 |               0 |    100.00 |
> |         1.0 |         4,999 |               1 |     99.98 |
> |         1.2 |         4,971 |              29 |     99.42 |
> |         1.5 |         4,423 |             577 |     88.46 |
>
> Only when pushing temperature to above 1.2, we see an increasing amount of failed generations. We will add a sentence to the paper clarifying that invalid generations are rarely observed at our default sampling settings and that we include the lightweight filter simply to make the pipeline robust.
>
> **line 280. "l points are sampled using a temperature parameter of 0.7 unless otherwise specified.". For the two problems you use two different temperature hyperparameters. How are these tuned, and is the performance very sensitive to this choice?**
>
> We did not aggressively tune temperature for our experiments, we selected the temperatures according to our past experience for the DB task, as T=0.7 is commonly used for LLM sampling. For the peptide task, we chose a slightly higher temperature to increase the diversity of the generated samples as we are generating 1000 samples instead of 50, as compared to the DB task. Here, we include additional abalations on the sampling temperature of the models and verify the robustness of our selection.
>
> The performance results (lower is better) are summarized as follows for the DB task:
>
> | Temperature | GPT-4o-mini-0718 | Qwen2.5-7B |
> | ----------- | ---------- | ---------- |
> | 0.1         | 84.97      | 84.42      |
> | 0.3         | 65.88      | 69.14      |
> | 0.5         | 62.19      | 63.97      |
> | 0.7         | 61.54      | 62.04      |
> | 1.0         | 60.09      | 61.25      |
> | 1.2         | 59.78      |    \        |
>
> We acknowledge that selecting temperature = 1.2 slightly outperforms the others, and recognize that our initial selection of 0.7 was based on preliminary studies conducted much earlier. Although temperature = 1.2 demonstrates marginal improvement, the 0.7 setting remains close to optimal and does not significantly affect our reported results.
>
> For peptides, we used T=1.0 to reduce repetition and increase useful diversity at initialization, consistent with the task's $\ge 75\%$ similarity constraint to the seeds. The analysis below (1000 samples per seed; 20 seeds per temperature; $\ge 75\%$ similarity threshold) shows that T=1.0 maximizes the number of peptides that are both unique and within constraint, providing the best tradeoff between novelty and staying in-distribution.
>
> |    Temp | unique fraction | within constraint | effective fraction | summed best MIC ↓ |
> | ------: | --------------: | ---------------: | -----------------: | ----------------------: |
> |     0.1 |         0.01620 |       **0.87340**|            0.01285 |                  155.91 |
> |     0.3 |         0.12315 |          0.85800 |            0.09085 |                  111.85 |
> |     0.5 |         0.33190 |          0.83030 |            0.24170 |                  103.94 |
> |     0.7 |         0.56620 |          0.78585 |            0.40035 |                   95.10 |
> | **1.0** |       0.83990   |        0.69090   |        **0.54350** |               **93.30** |
> |     1.2 |         0.93765 |          0.59700 |            0.53785 |                  100.67 |
> |     1.5 |     **0.98655** |          0.39660 |            0.38425 |                  103.56 |
>
> We can see from the table above that higher temperature leads to more diversity in the peptides generated, while also violating the seed similarity constraint more often. The temperature of 1.0 offers the best tradeoff, maximizing the pool of non-redundant, constraint-satisfying candidates and gives the best summed MIC over the test seeds. We will clarify in our manuscript that the peptide temperature was chosen heuristically for diversity and is now supported by this additional check.

---

> ### Author Response · Authors · 2025-11-21
> **Oracle budgets and color palettes**
>
> **line 257. I found it difficult to assess whether a budget of 200,000 oracle calls is reasonable in practice. You could consider adding a note on this.**
>
> **Figure 1. The labels in this figure are difficult to read, both due to the colours and the small font. Considering changing this.**
>
> Thanks for catching this! Sorry for this typo and apologiaze for the confusion, in all peptide experiments, we in fact run up to 20k oracle calls, not 200k. The "budget of 200,000 oracle calls" in the main text is a mislabel and will be corrected to 20,000 in the revised manuscript. We have checked the paper and this incorrect value appears only once in the main text and is correct in Appendix C.3. We will fix this typo and also update Figure 1 with larger fonts and a more readable and friendly color palette. For the budget itself, these evaluation budgets are consistent with or lower than evaluation budgets seen in other high dimensional Bayesian optimization literature. For example, the TuRBO paper have a budget of 20,000 evaluations for 60D rover trajectory planning, and the LOL-BO paper runs out to 120,000 oracle calls for the Ranolazine MPO task.
>
> Our budgets of 4,000 evaluations for DB and 20,000 for peptides are in line with prior work on high-dimensional Bayesian optimization in similarly challenging domains.
>
> 1. TurBO https://arxiv.org/abs/1910.01739
> 2. LOL-BO https://arxiv.org/abs/2201.11872

---

### Official Review · Reviewer_AGbs · 2025-11-01

**Soundness:** 3
**Presentation:** 3
**Contribution:** 3
**Rating:** 6
**Confidence:** 3

**Summary:**

This paper addresses the problem of scaling with multi-task Bayesian optimization and proposes the Bayesian Optimization with LLM Transfer (BOLT) method. BOLT is a simple approach that uses LLMs to generate candidates for new tasks, which provides strong initializations for BO. BOLT scales MTBO without saturation, effectively handling MTBO settings up to 1500 tasks.
The experiments on database query optimization and antimicrobial peptide design showed that BOLT is effective in providing high-quality initializations, which yield strong few-shot performance, outperforming LLM-based MTBO methods.

**Strengths:**

-  The proposed method BOLT is simple and modular, i.e., can be plugged into any BO loop.

- BOLT avoids saturation observed in common-shared GP methods and provides scaling.

- BOLT is empirically evaluated on diverse real-world use-cases, particularly, two high-throughput domains: database query optimization and antimicrobial peptide design.

**Weaknesses:**

-  BOLT requires a task description context that can be used in an LLM prompt to define the task. This excludes common MTBO settings.  Although the authors raise this point as a limitation, they still state BOLT as broadly scalable MTBO. A more precise scope statement or a counter-example domain would help to clarify this better.

- How sensitive is BOLT's performance to the outer loop? The ablations on top-k size for fine-tuning and fine-tuning frequency would have provided a better sensitivity analysis.

- It would strengthen the empirical effectiveness of BOLT better if the methods are analyzed under a fixed compute budget. That is, in terms of the compute budget, would BOLT still outperform if the total cross-task compute, including oracle evaluations and finetuning cost, is set to the same budget across methods?


- MINOR:

- - Line 64-65, the references for Optformer and LLAMBO should be added.
- - Line 457: "illustrates" should be corrected as "illustrate".

**Questions:**

- How would the performance of BOLT change if task contexts are perturbed? Could authors discuss robustness to context perfurbations? To test LLM's generalization performance vs. memorization?


- See also the Weaknesses above.

---

> ### Author Response · Authors · 2025-11-20
> **Clarifications on scope and sensitivity analysis**
>
> ### BOLT requires a task description context that can be used in an LLM prompt to define the task. This excludes common MTBO settings. Although the authors raise this point as a limitation, they still state BOLT as broadly scalable MTBO. A more precise scope statement or a counter-example domain would help to clarify this better.
>
> We completely agree that this is a limitation, and we point it out at beginning of section 3. Our method explicitly assumes (i) a shared input domain across tasks and (ii) the existence of a task description context $C[f_t]$ that can be fed into an LLM (e.g. SQL text for DB queries or a seed peptide sequence). This excludes common settings such as hyperparameter optimization across different datasets, where tasks differ mainly through training data and there is no concise, semantically meaningful text description. In the revised manuscript we will make this more prominent in the introduction and discussion, replacing wording like "broadly scalable MTBO" with a more precise statement that BOLT is designed for large collections of tasks that have informative textual (can be represented as strings) contexts.
>
> ### How sensitive is BOLT's performance to the outer loop? The ablations on top-k size for fine-tuning and fine-tuning frequency would have provided a better sensitivity analysis
>
> To check sensitivity of BOLT's performance to the outer loop, we ran three non‑overlapping BOLT‑200 (no self-augmentation) variants on the DB task, each fine‑tuned on 200 disjoint training queries with K=10. Their Best@50 summed runtimes are 79.67, 84.09, and 84.21 seconds, compared to 87.84 for BOLT‑50 and 82.31 for BOLT‑893 (both without self‑augmentation; Table 1). With two runs lie between the original BOLT-50 and BOLT-893 curves, with one slightly outperforming BOLT-893 (no self-augmentation). This pattern is consistent with Figure 1's scaling trend and suggests that the variability across BO traces exists but is moderate relative to the performance gap between small-T and large-T BOLT runs. We will add these 3xBOLT‑200 ablations to the appendix and clarify in the main text that we fine‑tune on top‑K points from individual BO runs.
>
> We also include an abalation on the specific choice of K. Keeping the same training set as the original BOLT‑893 run, we fine‑tuned four models using $K\in{1,2,5,20}$ top solutions per task. Their Best@50 summed runtimes are shown in the table below.
> | Top-K | Best@50 summed runtime (s) ↓ |
> |------:|------------------------------:|
> | 1     | 78.80                         |
> | 2     | 77.40                         |
> | 5     | 76.27                         |
> | **(original BOLT-893) 10** | **82.31** |
> | 20    | 69.56                         |
>
> This shows that (i) BOLT's performance improves as we increase the amount of high-quality data, and (ii) our original K=10 choice was rather on the conservative side rather than over-tuned, as larger K values showed better performance at the cost of more fine-tuning tokens (16 million tokens for K=20). We will report these results in the appendix and note in the main text that outer-loop hyperparameters such as K can shift performance by a few seconds but do not change the overall scaling behavior or the conclusions of Figure 1.

---

> ### Author Response · Authors · 2025-11-20
> **Runtime analysis**
>
> ### It would strengthen the empirical effectiveness of BOLT better if the methods are analyzed under a fixed compute budget. That is, in terms of the compute budget, would BOLT still outperform if the total cross-task compute, including oracle evaluations and finetuning cost, is set to the same budget across methods?
>
> This is a very good suggestion! We agree that comparing methods under a fixed compute budget is important for understanding the practical tradeoffs of BOLT. We have set up our experiments so that on every individual BO run, every method uses the same  oracle budget (4k for DB; 20k for peptides); BOLT only changes the initialization and adds offline LLM fine-tuning on training tasks. Below we provide a comparison of runtime and cost across BOLT and baseline MTBO methods:
>
> #### 1. STBO
>
> Each STBO run for the database (DB) tasks required approximately 15–20 GPU hours on a single NVIDIA RTX A5000/A6000 GPU, averaging roughly 18 hours per task. Given that we conducted around 1500 tasks, the total GPU time consumed by STBO runs alone was about:
>
> $$
> 1436 \text{ tasks} \times 15\text{–}20 \text{ GPU hours/task} \approx \textbf{25,000 GPU hours}
> $$
>
> #### 2. Baseline MTBO
>
> For comparison, for the baseline MTBO methods:
>
> * **SGPE and POGPE:** Significantly slower, each run exceeded 100 GPU hours per task and could not reach 1000 oracle calls within a reasonable time frame. We terminated these methods at 110 GPU hours per run.
> * **DKT/FSBO (10/20/50):** Computationally on par with STBO at 15–20 GPU hours per task, the added feature extractors etc add negligible extra overhead compared to training the GP alone on the same amount of data.
>
> Running all 1436 DB tasks using SGPE and POGPE would thus incur far greater costs than STBO alone, because SGPE and POGPE scale quadractically in the number of tasks. DKT/FSBO have similar computation costs to STBO, but don't improve much beyond 50 tasks or so.
>
> #### 3. BOLT Computational Costs
>
> #### Fine-Tuning Costs:
>
> * **GPT-4o-mini (OpenAI API):** Fine-tuning for 2 epochs on the largest BOLT dataset (BOLT-1426) used 26 million tokens, corresponding to a modest cost of approximately 78 usd. Taking into account fine-tuning across all BOLT datasets (BOLT-893, BOLT-1138, and BOLT-1426), the total fine-tuning consumed 60 million tokens, costing around 180 usd.
>
> To contextualize, this cost equates to approximately 400 GPU hours if performed locally using on-demand NVIDIA RTX A6000 GPUs.
>
> * **Local Fine-Tuning (QWEN2.5-7B):** Fine-tuning locally using QWEN2.5-7B took approximately 32 GPU hours (using four RTX A6000 GPUs over 8 hours). For emphasis: this corresponds to adding about 1-2 additional tasks of runtime (so, say 1428 tasks instead of 1426).
>
> #### Inference Costs:
>
> * Drawing 50 initialization samples per task from the locally fine-tuned model took roughly 1 GPU-minute per task (24 GPU-hours total across all 1436 tasks), which is trivial compared to the STBO runs themselves.
>
> #### 4. Overall Cost Comparison
>
> We summarize the relative costs for the query plan optimization task in the table below:
>
>
>
> | Method                     | GPU Hours (per Task) | GPU Hours (50 Tasks) | GPU Hours (1436 Tasks)         | Additional Overhead            |
> | -------------------------- | -------------------- | -------------------- | ------------------------------ | ------------------------------ |
> | **STBO (Single-task BO)**  | 15–20                | \~900                | \~22,500–28,720                | —                              |
> | **SGPE/POGPE**             | >100 (capped at 110) | >5,000 (incomplete)  | Quadratic scaling (infeasible) | Very High                      |
> | **DKT/FSBO**               | 15–20                | \~900                | \~22,500–28,720 (estimated)    | Negligible (extra GP training) |
> | **BOLT + STBO** | —                    | \~964 (+7% vs STBO)       | \~22,750-28,970 (+1% vs STBO)               | ~1.1% of total BO runtime     |
>
>
>
> As we can see from the table, the small computational overhead of BOLT (fine-tuning plus inference) is heavily amortized across even just 50 tasks. Specifically, the incremental cost of fine-tuning and inference for BOLT adds **~1.1% additional GPU-time** compared to the substantial cost of the STBO baseline by 1436 tasks. BOLT introduces only a marginal computational overhead, making it quite practical, particularly when scaling to hundreds or thousands of tasks. While DKT/FSBO requires roughly the same per-task computation as STBO, SGPE/POGPE's computational requirement scales quadratically, quickly becoming infeasible for larger numbers of tasks. In contrast, BOLT's computational overhead is effectively linear (but with a much smaller constant than running more oracle calls), making scaling to 1436 tasks not only practical but highly advantageous. We will integrate this detailed computational resource comparison into our revised manuscript to clearly demonstrate the computational advantages of BOLT.

---

> ### Author Response · Authors · 2025-11-20
> **Context perturbation**
>
> #### How would the performance of BOLT change if task contexts are perturbed? Could authors discuss robustness to context perturbations? To test LLM's generalization performance vs. memorization?
>
>
> We agree that it is important to verify that BOLT is actually using the task description context rather than memorizing a set of good plans. To test this, we ran an additional context–plan shuffling experiment on the DB task: we took the fine‑tuned BOLT‑1426 model and randomly permuted the mapping between SQL query strings and the LLM‑generated plans, then evaluated best‑of‑50 summed runtime on the validation queries. Under this “context‑perturbed” setting, performance degraded to 402.61 s, compared to 61.54 s for the original BOLT‑1426 samples and 106.72 s for the BAO initialization baseline. Here, destroying the alignment between context and generated plans makes the LLM far worse than both its own unshuffled version and the strongest non-LLM initializer, indicating that BOLT’s gains come from context‑specific optimization behavior rather than simple memorization of a global set of good plans. We will describe this perturbation experiment in the revised manuscript.

---

### Author Response · Authors · 2025-12-03
**Summary of responses**

We thank the reviewers for their thoughtful and insightful comments during the rebuttal phase. Below, we summarize the concerns addressed, and additional results/experiments added during the discussion.

- **Sensitivity of the outer loop and training samples (Reviewer AGbs and Yjw8)**: We added three non‑overlapping BOLT‑200 runs showing modest variability relative to the large gains from scaling to BOLT‑893/1426, and a Top‑K ablation (K ∈ {1,2,5,10,20}) showing that more high‑quality points per task generally helps.
- **Compute budget and fairness of comparison (Reviewer AGbs)**: We quantified end‑to‑end GPU hours. STBO alone costs ~25k GPU‑hours on DB, while BOLT’s fine‑tuning + sampling adds only ~1% extra GPU time on all tasks. SGPE/POGPE are far more expensive (quadratic in #tasks), and DKT/FSBO is comparable to STBO but saturates after tens of tasks.
- **Robustness, context usage, and random/local search baselines (Reviewer AGbs, Yjw8, PwEA)**: We ran a context shuffling experiment, where performance collapsed to much worse than both unshuffled BOLT and BAO. We also added several random/local search baselines, all underperform BOLT‑initialized BO.
- **Global vs local BO once BOLT is available (Reviewer PwEA)**: We implemented a recent “vanilla” global BO method on the DB task and compared it to our local BO method, both with BAO and with BOLT‑1426 initialization. BOLT‑initialization improves both global and local BO.
- **Sampling temperature and invalid generations (Reviewer KGBF)**: We reported temperature ablations for both domains, showing that our chosen temperatures (0.7 for DB, 1.0 for peptides) are near‑optimal and that invalid generations are very rare at these settings.
- **Reproducibility and missing implementation details (Reviewer Yjw8)**: We released an anonymous GitHub repo with full BO + BOLT code, and filled in missing training details.



We hope the above overview will provide a concise way to navigate through the information on this page.

---

### Meta-Review · Area_Chair_k9Mq · 2025-12-10

**Summary:**

Overall, this paper was well received. The reviewers saw the value in the proposed method to use LLMs to transfer knowledge from previous tasks to optimize a new task.

While they had multiple questions, no specific concern seemed to be of high importance across all reviewers. Nevertheless, questions were asked around how the method benefits the multi-task BO setting, and asking for more details on the trained VAE and more clearly defining the limitations of the method (needs a textual representation of the task for context) as well as estimating the variability of the method across various hyper-parameters (e.g. temperature, budget, K).

**Reviewer Concerns:**

The authors' responses were thorough and multiple experiments were performed to answer the requests of the reviewers. The more contentious point I found was that of reviewer PwEA, suggesting that a strong initialisation was sufficient to obtain good performance. The authors have provided multiple experiments and a careful response to this point that I found convincing.

**Reviewer Scores:**

Three reviewers leaned towards acceptance before the discussion, and I believe they would have kept their positive score. The only reviewer with a less favourable score had comments around the structure of the paper, the absence of code in the original submission, details of the VAE and an additional random search baseline from the VAE. I found that some of these comments were relatively minor (1-3) and the authors provided additional experiments and code, so I think the reviewer would not have opposed publication.

---

### Decision · Program_Chairs · 2026-01-26

Accept (Poster)